# Host cell CRISPR genomics and modelling reveal shared metabolic vulnerabilities in the intracellular development of *Plasmodium falciparum* and related hemoparasites

Marina Maurizio[1,7], Maria Masid [2,3,7], Kerry Woods[1], Reto Caldelari[4], John G. Doench [5], Arunasalam Naguleswaran [1], Denis Joly[3], Martín González-Fernández [1], Jonas Zemp[1,4], Mélanie Borteele[3], Vassily Hatzimanikatis [3], Volker Heussler [4], Sven Rottenberg[1] ✉ & Philipp Olias [1,6] ✉

Parasitic diseases, particularly malaria (caused by *Plasmodium falciparum*) and theileriosis (caused by *Theileria spp.*), profoundly impact global health and the socioeconomic well-being of lower-income countries. Despite recent advances, identifying host metabolic proteins essential for these auxotrophic pathogens remains challenging. Here, we generate a novel metabolic model of human hepatocytes infected with *P. falciparum* and integrate it with a genome-wide CRISPR knockout screen targeting *Theileria*-infected cells to pinpoint shared vulnerabilities. We identify key host metabolic enzymes critical for the intracellular survival of both of these lethal hemoparasites. Remarkably, among the metabolic proteins identified by our synergistic approach, we find that host purine and heme biosynthetic enzymes are essential for the intracellular survival of *P. falciparum* and *Theileria*, while other host enzymes are only essential under certain metabolic conditions, highlighting *P. falciparum*'s adaptability and ability to scavenge nutrients selectively. Unexpectedly, host porphyrins emerge as being essential for both parasites. The shared vulnerabilities open new avenues for developing more effective therapies against these debilitating diseases, with the potential for broader applicability in combating apicomplexan infections.

Apicomplexa are unicellular parasites that can cause severe and life-threatening diseases in humans and other animals. The malaria-causing *Plasmodium falciparum* (*P. falciparum*) is the most impactful human parasite, with more than 245 million malaria cases and more than 600,000 deaths reported annually[1]. Other socio-economically significant apicomplexans include the related bovine parasites in the genus *Theileria*, which are responsible for the deaths of more than 1 million cattle annually, significantly impacting livestock productivity,

[1]Institute of Animal Pathology, Vetsuisse Faculty, University of Bern, Bern, Switzerland. [2]Ludwig Institute for Cancer Research, Department of Oncology, University of Lausanne and Lausanne University Teaching Hospital (CHUV), Lausanne, Switzerland. [3]Laboratory of Computational Systems Biotechnology, École Polytechnique Fédérale de Lausanne (EPFL), Lausanne, Switzerland. [4]Institute of Cell Biology, University of Bern, Bern, Switzerland. [5]Broad Institute of MIT and Harvard, Cambridge, MA, USA. [6]Institute of Veterinary Pathology, Justus Liebig University, Giessen, Germany. [7]These authors contributed equally: Marina Maurizio, Maria Masid. ✉e-mail: sven.rottenberg@unibe.ch; philipp.olias@vetmed.uni-giessen.de

especially in small-scale cattle farming in the Global South[2,3]. Notably, highly virulent *Theileria* species have the unique ability among eukaryotic pathogens to induce cancer-like transformation of host cells, characterized by sustained proliferation and invasiveness[4–6].

A hallmark of apicomplexan parasites is their auxotrophy for multiple metabolites, meaning they lack the ability to synthesize essential nutrients and rely on their host cells. In recent years, our understanding of apicomplexan metabolism has significantly advanced, revealing significant variation in dependencies among different parasite genera[7,8]. However, due to the complex interactions with the host cell and the manipulation of key biological processes[9–11], there remains a lack of comprehensive understanding and detailed insight into the essential metabolic dependencies. We hypothesized that these auxotrophs, depending on their life cycle stage, rely on common host cell-derived metabolites for intracellular survival. Therefore, we explored the potential of targeting a single host metabolic pathway to block infection. Focusing on arguably the two most impactful genera globally, *Plasmodium* and *Theileria*, we investigated shared dependencies on key host metabolic proteins at the genomic scale (Fig. 1a). A common feature of both parasites is their initial expansion within nucleated host cells (the schizont phase) before massive expansion in red blood cells. This first expansion occurs within hepatocytes (*Plasmodium*) and white blood cells (*Theileria*), making it an ideal life cycle stage for comparative analysis.

Genome-wide screens, using tools such as CRISPR/Cas9 and RNAi, have proven to be highly effective in the identification of fitness-conferring genes of intracellular pathogens[12–14]. Only very few large-scale screens have focused on parasite-specific host gene essentiality, including the murine malaria parasite *Plasmodium yoelii* (*P. yoelii*), and the zoonotic parasites *Cryptosporidium parvum* and *Toxoplasma gondii*[15–17]. While *Theileria* schizonts can be studied in cell culture with quantitative proteomics and host genomic technologies, *Plasmodium* schizont-infected hepatocytes are technically difficult to study in large numbers, and genome-scale host metabolic data on the major malaria-causing species, *P. falciparum*, are lacking. Genome-scale metabolic models (GEMs) have emerged as invaluable tools in computational systems biology, providing a comprehensive and systems-level understanding of cellular metabolism[18]. By integrating genomic, biochemical, and physiological data, these models provide a holistic representation of the cellular metabolic networks[19]. In recent years, GEMs have been instrumental in investigating parasite metabolism[20,21], including that of apicomplexan parasites[10,22–25]. Advances in genome annotation, enzyme localization, and definition of available substrates have allowed for a more fine-tuned prediction of essential properties and metabolic activities within *Plasmodium*[14,26]. However, the complex interplay between host and parasite proteins is still poorly understood.

In this work, we reconstruct computational metabolic models to investigate the dependency of the *P. falciparum* liver stage on host metabolic genes. In parallel, we perform a CRISPR/Cas9 drop-out screen using a novel bovine genome-wide library in *Theileria annulata* (*T. annulata*) schizont-infected macrophages, the life cycle stage most analogous to the *P. falciparum* liver stage. Ultimately, we identify the core set of metabolic host genes essential for the survival of both *Plasmodium* and *Theileria* parasites.

## Results

### Modeling metabolic interactions between *Plasmodium falciparum* and the human hepatocyte

Aiming to understand the metabolic dependency of the malaria parasite *P. falciparum* during the liver stage, we developed a systems biology approach to explore how the parasite acquires nutrients from the host. Towards this, we first reconstructed a human hepatocyte-specific metabolic model (Fig. 1b) by collecting data from the Human Protein Atlas (www.proteinatlas.org)[27] and previous hepatocyte metabolic networks[28,29]. Then, we used this information and the human

Recon 3D[30,31] model to identify the metabolic reactions associated with genes expressed in liver cells and the additional reactions required to synthesize biomass building blocks (*Methods*). Secondly, we built a liver-specific metabolic model for *P. falciparum* (liver-iPfa) derived from the iPfa[26] GEM. This process involved determining nutrients available for the parasite based on metabolites present in the cytosol of the reconstructed hepatocyte model (*Methods*). The liver-iPfa model was used to assess the ability of *P. falciparum* to synthesize biomass and investigate its nutritional requirements (Fig. 1c). Additionally, we introduce the concept of the parasitosome, a metabolic reaction summarizing the metabolic interactions of an intracellular parasite within its host cell. The parasitosome reaction is derived from the metabolic reactions that the parasite uses to grow given certain nutritional conditions. In this reaction, substrates represent nutrients that the parasite takes up from the host's cytosol, and products represent metabolites that the parasite secretes into the host (Fig. 1d and *Methods*). Given the flexibility of the parasite's metabolism to adapt to varying environmental conditions, alternative parasitosomes may exist, representing different metabolic configurations of the parasite (Fig. 1d). Finally, the alternative parasitosomes are integrated into the hepatocyte model creating a host-parasite model that allows us to study the dependency of the *P. falciparum* schizont on the hepatocyte's metabolism (Fig. 1e). Using this approach, we investigated how the essentiality of host genes for parasite survival changes across different metabolic configurations of *P. falciparum*, identifying potential drug targets.

### In silico dependency of *Plasmodium falciparum* on the metabolism of the human hepatocyte

One must consider the parasite's ability to adapt its metabolism within the host cell to identify effective host gene targets for inhibiting parasite proliferation. For instance, *P. falciparum* demonstrates versatility by scavenging some nutrients from the host cytosol (as an auxotrophic parasite) or synthesizing them when unavailable (as a partial prototrophic parasite). This adaptability is crucial to understand when designing compounds that target host enzymes responsible for nutrient production. Therefore, we used the liver-iPfa model to investigate both scenarios: when *P. falciparum* operates in an auxotrophic mode or in a partial prototrophic mode (Fig. 2a). In the first case (auxotrophic), we simulated a parasite that has access to metabolites present in the cytosol of the hepatocyte model, resulting in 165 nutrients (Supplementary data 1 and *Methods*). In the second case (partial prototrophic), we simulated a parasite that only needs to consume the minimal number of metabolites from the hepatocyte cytosol for survival, resulting in a set of 47 nutrients (Fig. S1a, Supplementary data 1 and *Methods*). The reduction in nutrient requirements in the partial prototrophic case is explained by the fact that the parasite synthesizes some of the metabolites that it takes up from the host in the auxotrophic case. For example, the simulated auxotrophic parasite takes up most of the amino acids, saccharides, fatty acids, and nucleotides from the host (Figs. 2b, S1b). Conversely, when we simulate *P. falciparum* in a partially prototrophic mode it relies on its own metabolism and engages intracellular reactions from central carbon pathways, fatty acid metabolism, pyrimidine metabolism, and vitamin metabolism to synthesize these compounds (Figs. 2c, S1c). This results in elevated levels of by-products such as fatty acids and detoxification molecules secreted into the hepatocyte's cytosol (Figs. 2b, S1b). Overall, using the novel concept of parasitosomes, we have uncovered a total of 151 alternative metabolic states that an auxotrophic *P. falciparum* can adopt when growing inside the hepatocyte, along with 8 possible metabolic states that it adopts when operating as a partial prototrophic parasite.

The reconstructed models were then used to investigate which hepatocyte metabolic genes are essential for *P. falciparum* proliferation but dispensable for hepatocyte survival. To this end, we

performed in silico gene essentiality analyses in both the uninfected hepatocyte model and the hepatocyte-*P. falciparum* integrated model, considering the different metabolic states of the parasite (Fig. 2d). Computational knockout of each of the 1927 metabolic genes in the

hepatocyte model identified 55 essential genes for the healthy hepatocyte (Fig. S3a). Subsequently, in silico knockout simulations in the integrated model revealed 209 hepatocyte genes essential for at least one parasitosome in the case of auxotrophic parasite simulation,

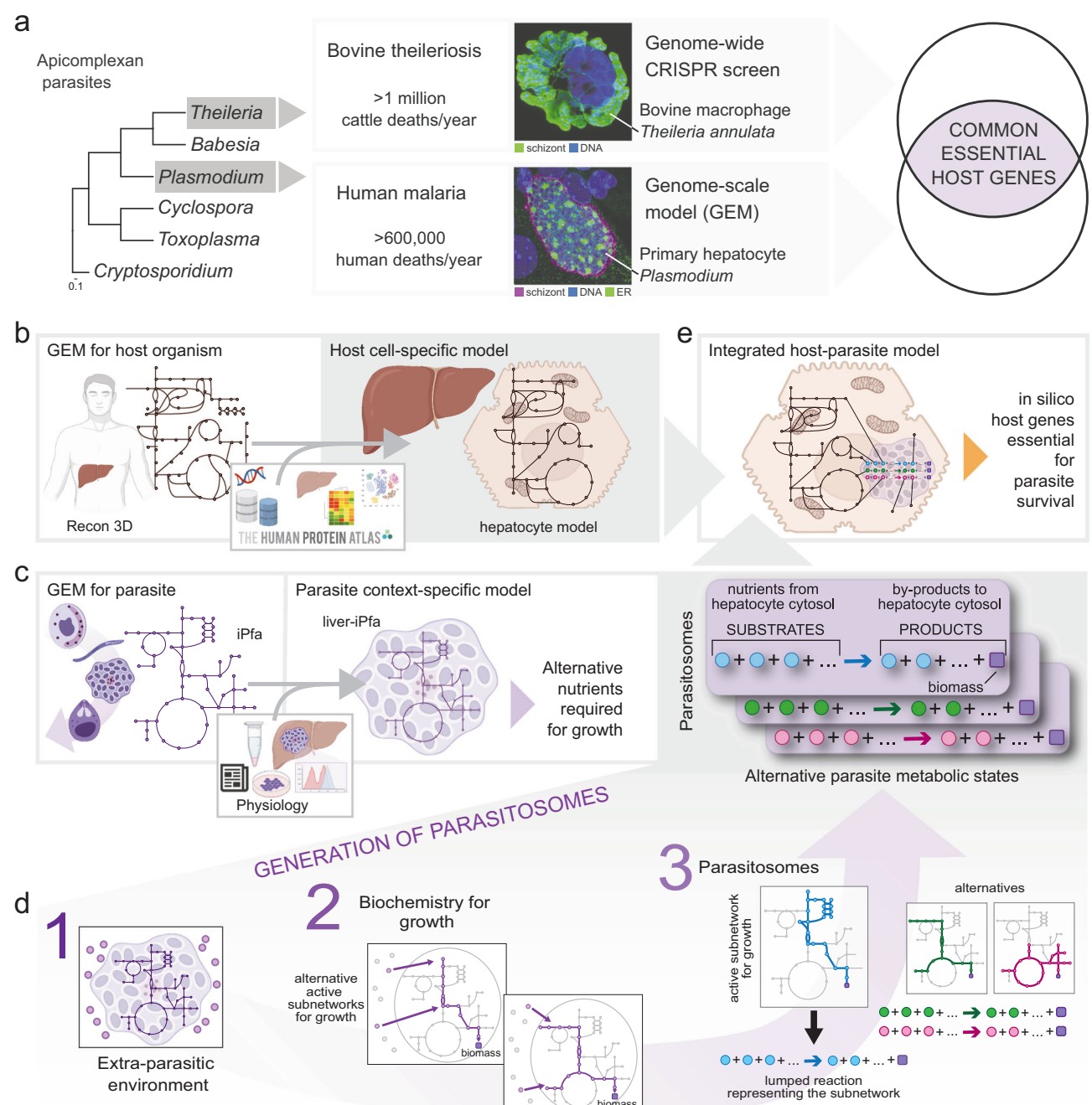

**Fig. 1 | Workflow to study host-parasite metabolic interactions. a** Schematic representation of the analysis to identify common drug targets for apicomplexan parasites. *Theileria* and *Plasmodium* are phylogenetically close within the Apicomplexa phylum and they are major pathogens causing life-threatening theileriosis in cattle and malaria in humans. Both parasites exhibit a multinucleated schizont stage (green and magenta) that develops within a mononucleated host cell (blue), with generation of numerous new parasites. Our multidisciplinary approach implies the utilization of state-of-art methodologies, such as genome-scale modeling for *Plasmodium*-infected hepatocytes and in vitro CRISPR screening for *Theileria*-infected macrophages, to identify shared essential metabolic genes for parasite development and survival. **b** Reconstruction of a hepatocyte-specific model from the human genome-scale metabolic model by integrating *omics* data. **c** Reconstruction of a liver-specific *P. falciparum* metabolic model from the iPfa

genome-scale model by restricting the extra-parasitic environment to metabolites available in the cytosol of the liver metabolic model. The reconstructed liver-iPfa model is used to analyze the nutritional requirements of the parasites and to formulate parasitosome reactions representing different metabolic states that the parasites can acquire to adapt to their environment and synthesize biomass. **d** Scheme of the computational workflow to generate parasitosomes: 1. Define the extra-parasitic environment; 2. Identify the reactions required for growth under the given conditions, and alternatives; 3. Extract the corresponding parasitosome reactions by collapsing the reactions identified in step 2 into one reaction containing the nutrients and products that each parasite state takes up and secretes into the host cytosol. **e** Reconstruction of an integrated host-parasite model, by incorporating the parasitosome reactions into the liver metabolic model.

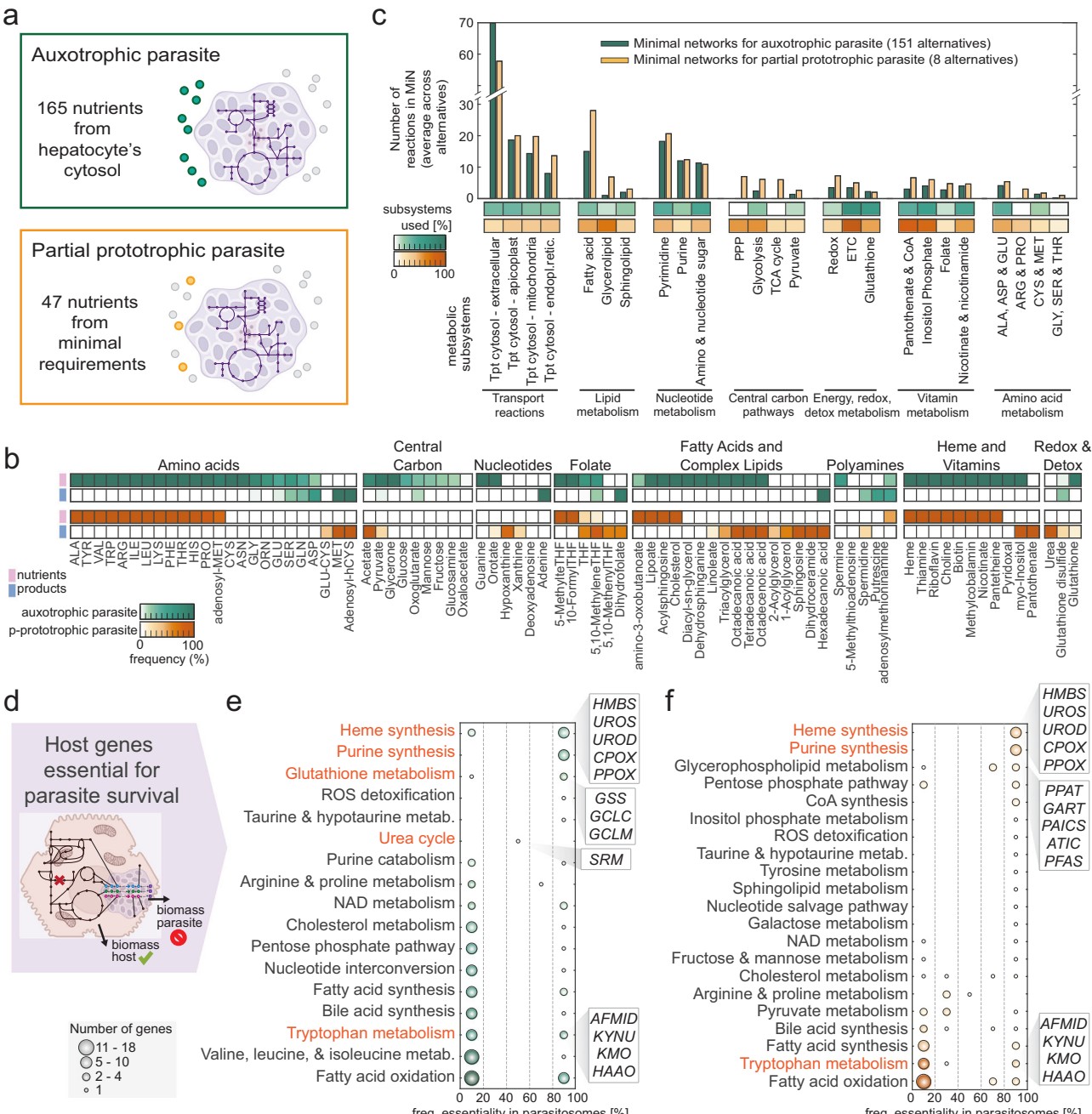

**Fig. 2 | Analysis of *P. falciparum* metabolic states and host genes essential for parasite growth. a** Case study I – auxotrophic *P. falciparum* with access to the entire hepatocyte's cytosol metabolome. Case study II – partial prototrophic *P. falciparum* uptaking a minimal set of nutrients from the hepatocyte's cytosol. **b** Metabolic composition of the alternative parasitosomes in case study I (auxotrophic parasite) and II (partial prototrophic parasite). Parasitosomes are representations of the nutrients taken up from and products secreted to the hepatocyte by parasites in different metabolic states. **c** Assignment to metabolic subsystems of the reactions that compose the minimal networks for the different parasitosomes, when considering auxotrophic (green) or partial prototrophic (orange) metabolic configurations of *P. falciparum*, in terms of number of reactions and percentage of subsystem coverage for selected pathways. **d** Scheme of gene essentiality analysis performed in the integrated host-parasite model. **e, f** Hepatocyte genes essential for *P. falciparum* in case study I (auxotrophic state, green) and in case study II (partial prototrophic state, orange) for selected pathways. Classification by associated metabolic processes, number of genes, and frequency of essentiality across parasitosomes.

spanning 76 metabolic pathways including heme synthesis, purine synthesis, glutathione metabolism, urea cycle, tryptophan metabolism, and fatty acid oxidation (Figs. 2e, S2a, Supplementary data 2). In the case of *P. falciparum* growing in a partially prototrophic state within the hepatocyte, we identified 110 genes essential for at least one parasitosome, belonging to 57 pathways including heme synthesis, purine metabolism, pentose phosphate pathway, and fatty acid metabolism (Figs. 2f, S2b, Supplementary data 2). The higher

essentiality observed in the case of auxotrophic parasites stems from their greater dependence on host metabolism compared to partial prototrophic parasites, which utilize their own metabolism for survival.

The different metabolic states in which the parasite can exist within the hepatocyte results in varying host gene essentiality, highlighting the flexible nature of *P. falciparum*'s interactions with the hepatocyte (Fig. S3b, c). This adaptability is crucial to consider in drug

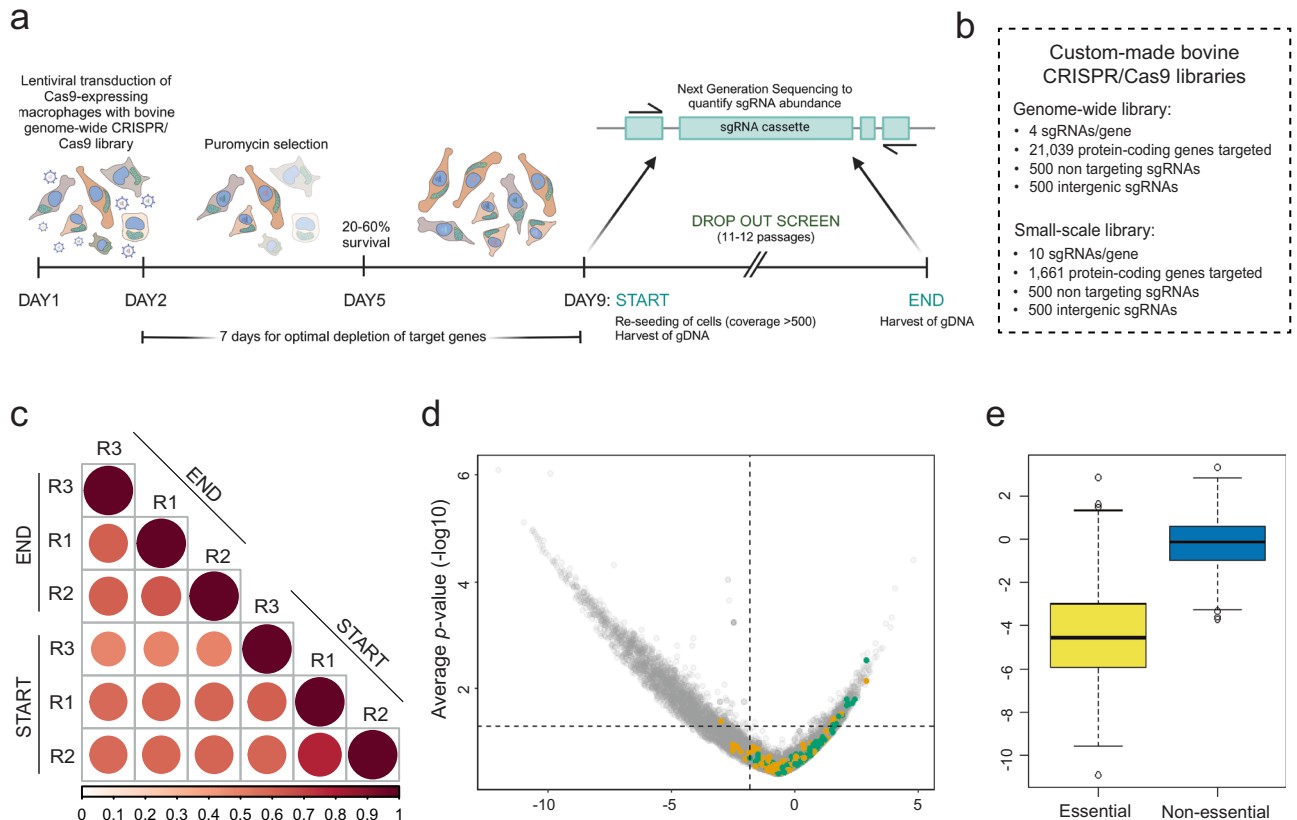

**Fig. 3 | Genome-wide CRISPR/Cas9 screen in *Theileria annulata*-infected bovine macrophages identify host factors essential for parasite survival. a** Schematic workflow of CRISPR dropout screens. Cas9-expressing infected macrophages (TaC12) were transduced with a pooled CRISPR sgRNA lentiviral library and maintained in culture for at least 11 passages. Genomic DNA was harvested at the START and END points and sgRNA abundance was quantified. The screen was performed in 3 biological replicates. **b** Detailed information on the bovine CRISPR/Cas9 libraries used in this study. **c** Correlation plot screen replicate. Raw counts of each replicate were used as dataset. Dark red indicates high similarity between replicates. **d** Volcano plot showing the hypergeometric distribution analysis of the TaC12 genome-wide screen. Each gene is plotted based on its average log$_2$ fold change (LFC, z-normalized using control dispersion) and the negative log$_{10}$ of its *p*-value.

Intergenic and non-targeting controls are shown in orange and green, respectively. The horizontal dotted line corresponds to $p = 0.05$, and the vertical dotted line corresponds to average z-normalized LFC of the controls (intergenic and non-targeting) minus 1x standard deviation of the whole genetic screen. One-sided Fisher's exact test was performed as statistical test. **e** Boxplot representing TaC12 genome-wide screen scoring (z-normalized LFC values) of groups of annotated "essential" and "non-essential" genes. The data presented are average values from 3 biological replicates (constituting three independent transductions performed on different days). The black line denotes the median value (50th percentile), while the lower and upper bounds of the box represent the 25th to 75th percentiles. The lower and upper whiskers represent the 5th and 95th percentiles, respectively.

target discovery. For instance, we examined the in silico interactions between hepatocytes and parasites for polyamines (Fig. S1d). Our findings reveal that spermidine synthase (*SRM*) is essential when *P. falciparum* is modeled to primarily take up spermine, adenosylmethionine (SAM) and 5-methylthioadenosine (5MTA) from the hepatocyte and to secrete decarboxylated adenosylmethionine (dcSAM), thus relying on hepatocyte polyamine synthesis to produce spermine and eliminate parasite by-products (Fig. S1e, f). However, when *P. falciparum* is simulated to synthesize spermine from ornithine using its own polyamine synthesis, host *SRM* is not essential as the parasitosomes preferentially transport ornithine from the hepatocyte, synthesize spermine, and secrete 5MTA into the hepatocyte (Fig. S1e, f).

In total, we computationally identified 19 host genes that are essential for the survival of *P. falciparum* under all metabolic configurations (Fig. S3d). These 19 genes are the result of overlapping the genes predicted to be essential for all auxotrophic and partial prototrophic parasitosomes. Among them, we found genes from the heme synthesis pathway (*HMBS, UROS, UROD, CPOX,* and *PPOX*), purine synthesis pathway (*PPAT, GART, PAICS, ATIC,* and *PFAS*) and tryptophan metabolism (*AFMID, KYNU, KMO,* and *HAAO*). The other genes found are involved in NAD metabolism (*QPRT*), taurine and hypotaurine

metabolism (*CDO1*), pentose phosphate pathway (*RPIA*), extracellular transport (*SLC11A2*), and cholesterol metabolism (*SOAT1*).

## Genome-wide CRISPR/Cas9 in *Theileria annulata* schizont infected host cells identifies parasite-specific essential host genes

Having identified essential host genes for survival of *P. falciparum*, we next aimed at finding common host factors essential for *T. annulata*. While the generation of the hepatocyte-*P. falciparum* metabolic model was made possible by recent advances in modeling resources[26–31], *omics* data for bovine leukocytes and *Theileria* parasites remain limited and incomplete. We performed a comparative RNA sequencing of bovine macrophages following infection with *T. annulata*, and found that numerous metabolic genes, including some involved in purine and pyrimidine synthesis and fatty acid synthesis, are deregulated in *T. annulata*-infected cells (Fig. S4). The continuous proliferation of *T. annulata*-transformed cells makes them ideal for a genome-wide CRISPR/Cas9 dropout screen (Fig. 3a). Survival and growth of the transformed cell depends upon the presence of a viable parasite[32], and so loss of host cell viability will in many cases indicate a loss of parasite survival. To identify essential host factors, we generated a CRISPR sgRNA library based on the bovine ARS-UCD1.2 genome (Fig. 3b).

We expressed Flag-Cas9 in infected (TaC12) macrophages, and validated protein expression by Western blotting (Fig. S5a). We also used a fluorescence based Cas9 reporter assay to select for cells expressing active Cas9 (Figs. S5b, c, S11). The genome-wide screen was performed at a coverage where each sgRNA is expressed in at least 500 cells and in three biological replicates (Figs. 3c, S5d, e). Cells were transduced with lentiviruses for 24 h (Fig. 3a) and selected with puromycin. The dropout screen continued for at least 11 passages (4 weeks), with cells kept under selection and reseeded after each passage. Log$_2$ fold change (LFC) values were generated based on how sgRNAs changed in abundance throughout the screen after Illumina sequencing. The data were analyzed using a hypergeometric distribution, where gene LFC values were normalized using intergenic and non-targeting controls (Supplementary data 3) and displayed in a volcano plot (Fig. 3d). A hit with a LFC < 0 indicates a reduction in cell fitness upon gene inactivation and consequently these cells are less abundant at the end of the screen compared to the beginning. We assessed the functionality of our library using a reference set of essential and non-essential genes[33] (Fig. 3e). We then selected a list of ~1500 genes that significantly reduced cell fitness upon Cas9-mediated inactivation to design a CRISPR sgRNA sublibrary (Fig. 3b) which we used to perform a validation screen including an uninfected bovine macrophage cell line (BoMac) as control (Fig. 3a, Fig. S5f–i, Supplementary data 3). BoMac cells, derived from a bovine monocyte/macrophage lineage, represent the most effective system currently available for assessing the essentiality of bovine host genes within a macrophage background. Principal component analysis of log-normalized counts confirmed the similarity of DAY 0 samples for both TaC12 and BoMac cell lines, resemblance to the plasmid DNA library and a divergent behavior of these two cell lines upon prolonged time in culture (Fig. 4a). BoMac screens showed a lower replicate reproducibility, indicating a less consistent response in repeated experiments (Fig. 4b). About 80% of the ~1500 candidate genes identified in TaC12 cells were validated as crucial for parasite survival (Fig. S5f). By removing the common fitness-conferring genes identified in BoMacs (Fig. S5h), 653 host genes were defined as the exclusive *Theileria*-infected macrophage essentialome, henceforth referred to as the *Theileria* essentialome (Fig. 4c, Supplementary data 4).

### Host genes involved in proliferation, cell growth and metabolism are fitness-conferring in *Theileria*-infected bovine macrophages

To scrutinize the *Theileria* essentialome we performed a Gene Ontology analysis (Supplementary data 4). This revealed a significant enrichment of proteins involved in ribosome biogenesis and assembly (GO:0042254), mRNA metabolism (GO:0016071), RNA processing (GO:0006396), translation (GO:0006412) and cell cycle progression (GO:0022402), highlighting the importance of continuous protein synthesis to cope with the proliferative phenotype induced by *T. annulata*. Genes involved in the host cell metabolism make up 15% of the *Theileria* essentialome (Fig. 4d), including biosynthesis of isoprenoids (GO:0008299), glycerolipids (GO:0045017), polyamines (GO:0006596), glutathione (GO:0006749), heme (GO:0006783) and ubiquinone (GO:0006744). The tricarboxylic acid cycle is the most enriched metabolic process in our dataset according to KEGG pathway analysis (Fig. S6a, b). As the entire phylum of apicomplexan parasites lacks the de novo purine biosynthesis pathway, the *Theileria* essentialome contains several host genes involved in the biosynthesis of purine precursors (GO:0009167) (Figs. 4d, S7a). While enzymes that convert IMP to AMP are present (ADSS and ADSL), those deputed to synthesize GMP (IMPDH and GMPS) are absent, highlighting a specific requirement for adenine nucleotides by infected cells (Fig. S7b).

In both genome-wide and small-scale screens, perturbation of glutathione synthetase (*GSS*) and glutamate-cysteine ligase catalytic subunit (*GCLC*) significantly reduced the survival of infected

macrophages over time, while uninfected cells showed no defect in cell survival (Fig. 5a). A single CRISPR/Cas9 knockout of bovine *GSS* (Fig. 5a, Fig. S8a), resulted in a population of BoMac with 58.7% frameshift mutations in the target gene locus, while >95% of TaC12 retained the wild-type *GSS* sequence. This result shows that TaC12 *GSS* mutants are not viable and undergo cell death upon gene knockout. To support these data, we treated cells for 72 h with different concentrations of buthionine sulfoximine (BSO), a well-established irreversible inhibitor of γ-GCS[34], and observed that TaC12 were more sensitive than BoMac in the 30–1000 μM BSO range (Figs. 5a, S8c), confirming the importance of glutathione biosynthesis exclusively for infected cells.

Mammalian cells have a classical forward-directed polyamine biosynthetic pathway that produces polyamines from arginine and methionine, but they also can convert spermine back to spermidine and putrescine via a retroconversion pathway (Fig. 5b). One of the top depleted genes in the TaC12 screen was *SRM*, which was dispensable for BoMac (Fig. 5b). Targeted *SRM* CRISPR mediated knockout resulted in >82% frameshift mutations in BoMacs, while we were unable to knock out SRM in populations of TaC12 cells (Figs. 5b, S8a). Consistent with this finding we observed that when cells were treated with the nitric oxide donor sodium nitroprusside (SNP), a potent inhibitor of polyamine biosynthesis[35], TaC12 showed over 70% reduction in cell viability compared to BoMac (Figs. 5b, S8d).

Ablation of five out of the eight enzymes of the host heme biosynthetic pathway negatively affected the fitness of TaC12 cells in the screen (Fig. 5c), while no phenotype was observed in uninfected macrophages (Fig. 5c). We selected hydroxymethylbilane synthase (HMBS) for further validation, and we observed that no viable gene knockout could be achieved in *Theileria*-infected cells, whereas we were able to generate a polyclonal population of BoMac cells that acquired frameshift mutations in the target gene (Figs. 5c, S8a). Treatment with salicylic acid (SA), an inhibitor of ferrochelatase (FECH)[36], reduced the viability of TaC12 cells by >90%, whereas BoMac viability remained unchanged (Figs. 5c, S8e). Together, these data further validated the critical role of the host glutathione, polyamine, and heme biosynthesis in *Theileria*-infected cells.

### *Plasmodium* and *Theileria* schizont survival depend on a core set of shared host enzymes

*P. falciparum* liver-stage infection remains one of the least characterized but most attractive targets for malaria prevention as it represents a bottleneck in parasite development. Experimental limitations, such as the low infection rate and strict biosafety requirements, necessitate the use of metabolic modeling to study the host interplay of the hepatic schizont stages of the human malaria parasite. Conversely, *T. annulata* is well suited for in vitro culturing, as demonstrated by successful CRISPR screening in *Theileria*-infected macrophages. We investigated whether these pathogens rely on the same host metabolic pathways. We identified 99 metabolic enzymes in our *Theileria* essentialome and we searched for overlaps with the predicted host gene essentiality for liver stage *P. falciparum* in the auxotrophic and partial prototrophic scenarios. To this end, we identified common host genes that are essential for at least one of the alternative metabolic configurations simulated for liver stage *P. falciparum*, as well as the list of common host genes essential for all simulated metabolic configurations of *P. falciparum* (Fig. 6). We identified 28 host genes that are commonly essential for *Theileria* and *Plasmodium* in at least one of the simulated metabolic configurations, including genes encoding mitochondrial complex II subunits (*SDHA, SDHB, SDHC, SDHD*) and genes involved in polyamine (*AMD1, SRM*) and glutathione (*GCLC, GSS*) biosynthesis (Fig. 6). Of these only seven genes, belonging to the heme and purine biosynthetic pathways, were identified as commonly essential for *Theileria* and all simulated metabolic configurations of *P. falciparum*. This suggests that when metabolites such as polyamines and glutathione (GSH) are available for uptake in the host cytosol,

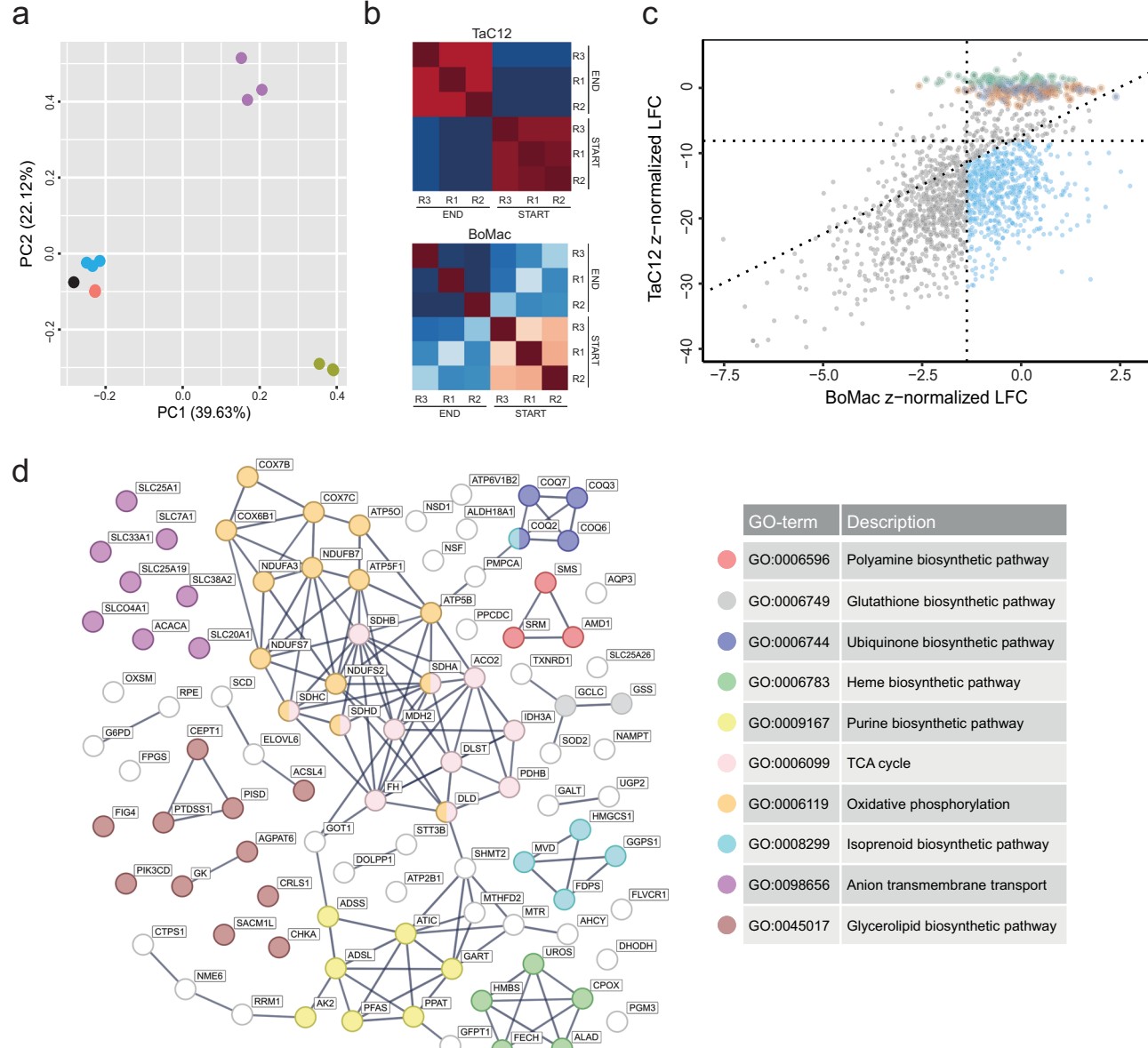

**Fig. 4 | Pathway analysis of fitness-conferring host genes for *Theileria* schizonts. a** Principal component analysis plot of the log-norm counts distribution of each biological replicate of small-scale screens. Color legend: BoMac_DAY0_r1-2-3 (blue), BoMac_END_r1-2-3 (purple), TaC12_DAY0_r1-2-3 (red), TaC12_END_r1-2-3 (green), CRISPR sublibrary (black). **b** Correlation plots of small-scale screen replicates. Raw counts of each replicate were used as dataset. Dark red indicates high similarity between replicates, while dark blue indicates the lowest correlation value among replicates. **c** Scatter plot showing the results of small-scale screens. Each gene is plotted based on z-normalized LFC values in infected versus uninfected cells. Genes depleted only in TaC12 cells are colored in light blue. Intergenic and non-targeting controls are shown in dark blue and green, respectively. One hundred non-essential genes from the TaC12 genome-wide screen are highlighted in orange. Dotted lines correspond to the average z-normalized LFC of the controls (intergenic and non-targeting) minus 1x standard deviation of each genetic screen. **d** Protein interaction network and Gene Ontology analysis of metabolic genes present in the *Theileria*-infected macrophage essentialome. The list of 99 metabolic genes was analyzed using the STRING database (https://string-db.org/) on July 19th 2023. Line thickness indicates the strength of data support. Active interaction sources considered: text mining, experiments, databases, co-expression. Only interactions with the highest confidence score are shown. GO enrichment analysis identified 10 major host metabolic pathways.

*Plasmodium* parasites scavenge these nutrients. However, in the absence of these molecules, our model suggests that parasite de novo biosynthesis is sufficient to produce the essential metabolites. All apicomplexans have lost the ability to synthesize purine nucleotides endogenously and rely on the mammalian host as a source of purines[37]. In line with this we found that the host genes *ATIC*, *GART*, *PFAS* and *PPAT* are highly fitness-conferring for *Theileria* and predicted to be essential for all *P. falciparum* parasitosomes (Fig. 6). We also identified the heme biosynthesis pathway, represented by the host genes *CPOX*, *HMBS* and *UROS*, as essential for both parasites under all conditions.

The prediction that host HMBS is essential for *P. falciparum* was unexpected, since *Plasmodium* possesses the entire enzymatic machinery for heme biosynthesis (Fig. S9)[8].

## Perturbation of environmental heme pools blocks *Plasmodium* schizont development

Our results imply that the significance of the host hepatic heme pathway has not been thoroughly investigated for *P. falciparum*. This finding constitutes a hitherto overlooked vulnerability of the parasite and an intriguing drug target. *Plasmodium berghei* (*P. berghei*) heme

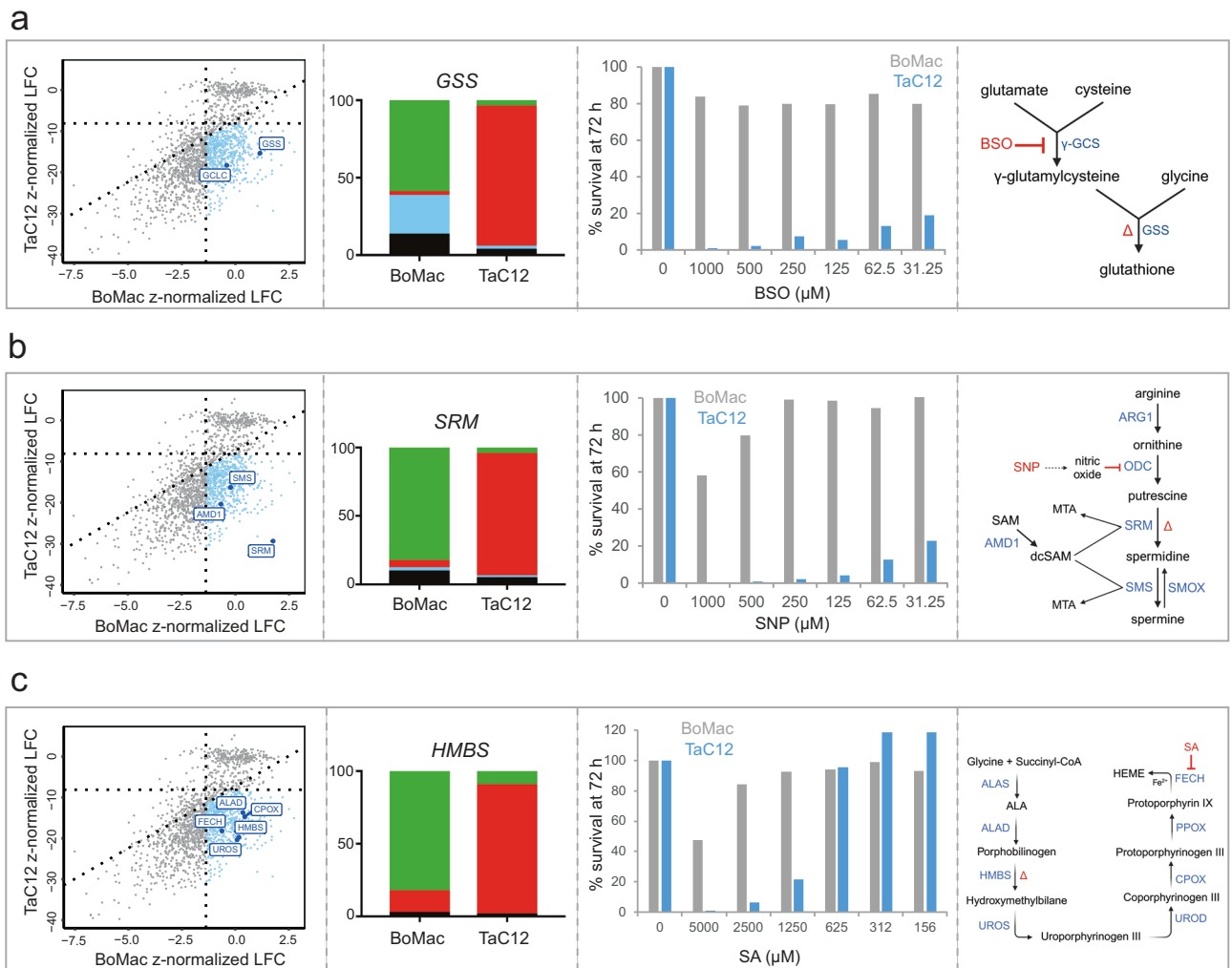

**Fig. 5 | Host glutathione, polyamine, and heme pathways are essential for *Theileria*. a–c** Scoring of *GCLC, GSS, AMD1, SMS, SRM, ALAD, HMBS, UROS, CPOX,* and *FECH* in TaC12 vs BoMac screens. TIDE analysis of *GSS, SRM,* and *HMBS* CRISPR/Cas9 knockout in BoMac and TaC12 cells (red, WT; green, frameshift mutation; light blue, in-frame mutation; black, not determined). The gene knocked out is indicated with delta (Δ) in the corresponding schematic biosynthetic pathway. Resazurin viability assay showing the percentage of survival of BoMac and TaC12 cells after 72 h treatment with three different enzymatic inhibitors: buthionine sulfoximine (BSO) (inhibitor of γ-GCS), sodium nitroprusside (SNP) (inhibitor of ODC), and salicylic acid (SA) (inhibitor of FECH). TIDE and viability experiments were performed in triplicate with similar results (Fig. S8a–e). A representative replicate is shown for each.

biosynthesis gene expression data indicate low activity of the endogenous pathway during biomass increase in the liver stage[38] (Fig. 7a). Interestingly, *Pb*FECH, the enzyme that catalyzes the final step of the pathway, is significantly upregulated, with expression peaking at the late exo-erythrocytic stage (54 and 60 hpi). Previous research has shown that *Pb*FECH knockout parasites grow to only half the size compared to wild-type parasites[39]. This suggests that parasites may use, at least in part, host cell-derived or systemically available porphyrins. Notably, our model and screening data suggest that the mammalian heme pathway plays a central role in the survival of the *Plasmodium* exo-erythrocytic stage, with three heme pathway genes being identified as essential for both *Plasmodium* and *Theileria*. To substantiate our results, we chose to knock out *HMBS*, which is upstream of UROS and CPOX (Fig. 5c), to inactivate the entire pathway (Figs. 7b, S8f). Clonal *HMBS*-deficient HAP1 cells do not grow in heme-depleted media, and this growth defect can be restored by hemin supplementation (Fig. 7c). We investigated whether *P. berghei* parasites can successfully develop in HAP1 cells with impaired heme metabolism. We infected six individual HAP1 Δ*HMBS* clones with *P. berghei* sporozoites and compared parasite size at 48 hpi. As predicted by our metabolic model, we observed a severe defect in intracellular

parasite growth, as parasites were significantly smaller compared to those in WT cells (Fig. 7d). The marked reduction in parasite size was common to all infected Δ*HMBS* clones. Next, we analyzed parasite growth after *HMBS* complementation. The reduced size phenotype observed in the parental knockout cells was fully rescued as shown in Fig. 7e for one biological replicate (see Fig. S12 for two further biological replicates), indicating that reactivation of the interrupted host heme biosynthetic pathway allows the successful growth of *Plasmodium* liver stage parasites.

## Discussion

Apicomplexan parasites, due to their inherent metabolic constraints, rely heavily on scavenging essential metabolites from their host cells[7,8,37]. By employing a synergistic approach that combines genome-scale metabolic network modeling of *P. falciparum* infected hepatocytes and genome-wide CRISPR/Cas9 screening of *T. annulata* infected macrophages, we have gained critical insights into the shared metabolic dependencies and pinpointed the host metabolic genes that are essential for parasite survival. We identified 28 host metabolic genes of the *Theileria*-infected macrophage essentialome shared with the predicted host gene essentiality of liver stage *P. falciparum*. Of these,

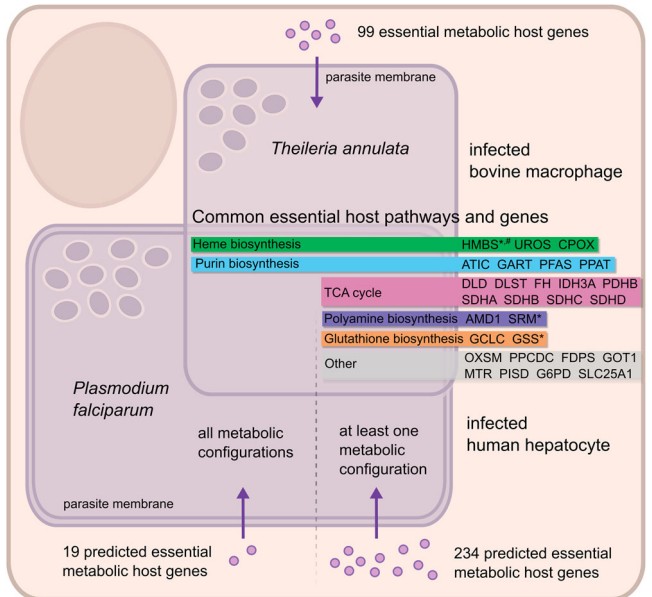

**Fig. 6 | Common host metabolic pathways and genes essential for *Plasmodium* and *Theileria* schizont survival in nucleated host cells.** Metabolic pathway association of 28 genes identified as the overlap of 99 metabolic host genes from *Theileria* essentialome and 234 host genes predicted to be essential for liver-stage *P. falciparum* in at least one of the parasitosomes, i.e., at least one simulated metabolic configuration of *P. falciparum* in the liver-stage. Seven genes essential for *Theileria* and *P. falciparum* in all metabolic configurations were characterized as the overlap of the 99 metabolically essential *Theileria* genes and the 19 host genes to be predicted as essential for all parasitosomes. Genes further validated experimentally in *T. annulata* (*) and *P. berghei* (#).

seven genes belonging to the heme biosynthesis and purine synthesis pathways were shown to be essential for *Plasmodium* growth under all modeled metabolic conditions. This highlights the need to consider the parasite's environment, including the host cell cytoplasm and the extracellular milieu surrounding the host cells when investigating host-parasite metabolic dependencies. Interestingly, we identified very little overlap with previously published CRISPR-screens, such as the study by Vijayan and colleagues[15] where they performed a genome-wide CRISPR screen in *P. yoelii* infected cells. While their experiment was designed to identify host factors important for early stages of liver stages and development (24 h post infection), our model focused on metabolic pathways essential for *Plasmodium* schizont development.

Our results show that *Plasmodium* schizonts preferentially rely on host glutathione but can switch to endogenous synthesis when the availability of metabolites in the host is limited. *Plasmodium* has an active GSH biosynthetic pathway that is vital for its development in the mosquito. Since *P. berghei GSS* knockout mutants develop efficiently in the liver in vivo[14], this supports our prediction of a preferential dependency on host *GSS*. While *Plasmodium* can potentially rely on its own glutathione synthesis pathway when needed, our data indicate that modulating host GSH levels, in combination with antimalarial therapy, may be a promising line of future research in malaria therapy. Targeting GSH in the context of anti-parasite therapy has parallels to anti-cancer therapy, where high GSH levels are also essential for cancer cell survival[40–42] and warrant exploration in the context of malaria and theileriosis therapies.

Diverse heme acquisition strategies have been described among Apicomplexan parasites. While *Plasmodium* and *Toxoplasma* possess a complete and functional de novo heme biosynthetic pathway, parasites such as *Babesia*, *Theileria* and *Cryptosporidium* depend on acquiring heme from the host cell[8]. Our experiments provided novel insights into the consequences of loss of the host heme pathway in *Plasmodium* and confirmed the unexpected predictions of our model.

When *HMBS* was compromised, *P. berghei* failed to grow normally. Strikingly, the complementation of HMBS completely reversed this phenotype. While there are conflicting data regarding the essentiality of *Pb*ALAS in the liver, external ALA could rescue *P. berghei* lacking ALAS[43,44]. Rathnapala et al. further demonstrated that FECH-deficient *P. berghei* grew smaller in size during liver stage development[39]. Taken together, this suggested that endogenous heme synthesis becomes critical during liver development, in contrast to the intraerythrocytic stages[43–46]. Our results show that the host heme synthesis enzymes upstream of FECH are highly fitness-conferring in liver stage *Plasmodium* parasites.

Host-directed therapies in host-parasite infections are of increasing scientific importance, as they offer promising therapeutic strategies to disrupt the parasite's life cycle, develop effective treatments, and mitigate drug resistance[47,48]. Our study identified crucial host proteins essential for the survival of two of the world's most lethal parasites, offering promising avenues for simultaneously targeting both host and parasite pathways.

## Methods

### Hepatocyte specific model

Starting with the thermodynamically curated human genome-scale Recon 3D[30,31], we reconstructed a hepatocyte metabolic model by taking into account the physiology of hepatocytes and the genes expressed in liver cells. Towards this end, we defined the physiology of hepatocytes by integrating in the human Recon3D model publicly available fluxomics data for 92 boundary reactions[28] and metabolomics data for 213 metabolites[29] from previous hepatocyte model reconstructions. Additionally, we set the growth rate of the hepatocyte to a maximum of 0.014 h⁻¹ corresponding to a doubling time of 49.5 h[49] and the ATP maintenance rate to at least 1.07 mmol/gDW/h[50]. The Human Protein Atlas (www.proteinatlas.org)[27] was used to identify 1853 metabolic genes present in liver cells. By mapping these genes to the Recon 3D reactions (using the gene-protein-reaction rules), 5926 reactions conformed the starting core of the hepatocyte model. We then formulated the following MILP to integrate additional reactions required for the core reactions to carry flux:

$$\textit{objective function} \quad \max \sum_{k=1}^{nc} z_k$$

subject to

$$\textit{FBA constraints} \quad \mathbf{S} \cdot \boldsymbol{v} = \mathbf{0}$$
$$\boldsymbol{v}_L \leq \boldsymbol{v} \leq \boldsymbol{v}_U$$

$$\textit{TFA constraints} \quad \triangle_r G_i' = \sum_{j=1}^{m} n_{i,j} \triangle_f G_j'^o + RT \ln\left(\prod_{j=1}^{m} x_j^{n_{i,j}}\right), \quad i=1,\ldots,n,$$
$$\triangle_r G_i' - M + M \cdot b_i^F \leq 0$$
$$-\triangle_r G_i' - M + M \cdot b_i^R \leq 0$$
$$v_i^{F,R} - M \cdot b_i^{F,R} \leq 0$$
$$b_i^F + b_i^R \leq 1$$

$$\textit{constraints for} \quad b_k^F + b_k^R + C \cdot z_k \leq C, \qquad k=1,\ldots,nc,$$
$$\textit{non-core reactions} \quad \boldsymbol{v}_c^F + \boldsymbol{v}_c^R \geq \epsilon, \qquad \forall c \in core$$

where $m$ is the number of metabolites in the network, $n$ is the number of reactions in the network, $nc$ is the number of non-core reactions, $z_k$ are binary variables for all the non-core reactions, $\mathbf{S}$ is the stoichiometric matrix, $\boldsymbol{v}$ are the net fluxes for all the reactions and $v_i^F$, $v_i^R$ are the corresponding net-forward and net-reverse fluxes, so that, $v_i = v_i^F - v_i^R$, for all $i=1,\ldots,n$. $\boldsymbol{v}_L$ and $\boldsymbol{v}_U$ are the lower and upper bound, respectively, for all the reactions in the network. $\Delta_r\mathbf{G}'$ is the Gibb's free energy of the reactions and $\mathbf{b}^F$ and $\mathbf{b}^R$ are the binary variables for the forward or reverse fluxes of all the reactions (coupled to TFA). $M$ is a big constant (bigger than all upper bounds), $C$ is an arbitrary large number, and $\epsilon$ a small number. With this formulation, if $z_k = 0$, then reaction $k$ is active.

As a result, the reconstructed hepatocyte model contains a total of 3899 metabolites, 6209 reactions, and 1927 genes.

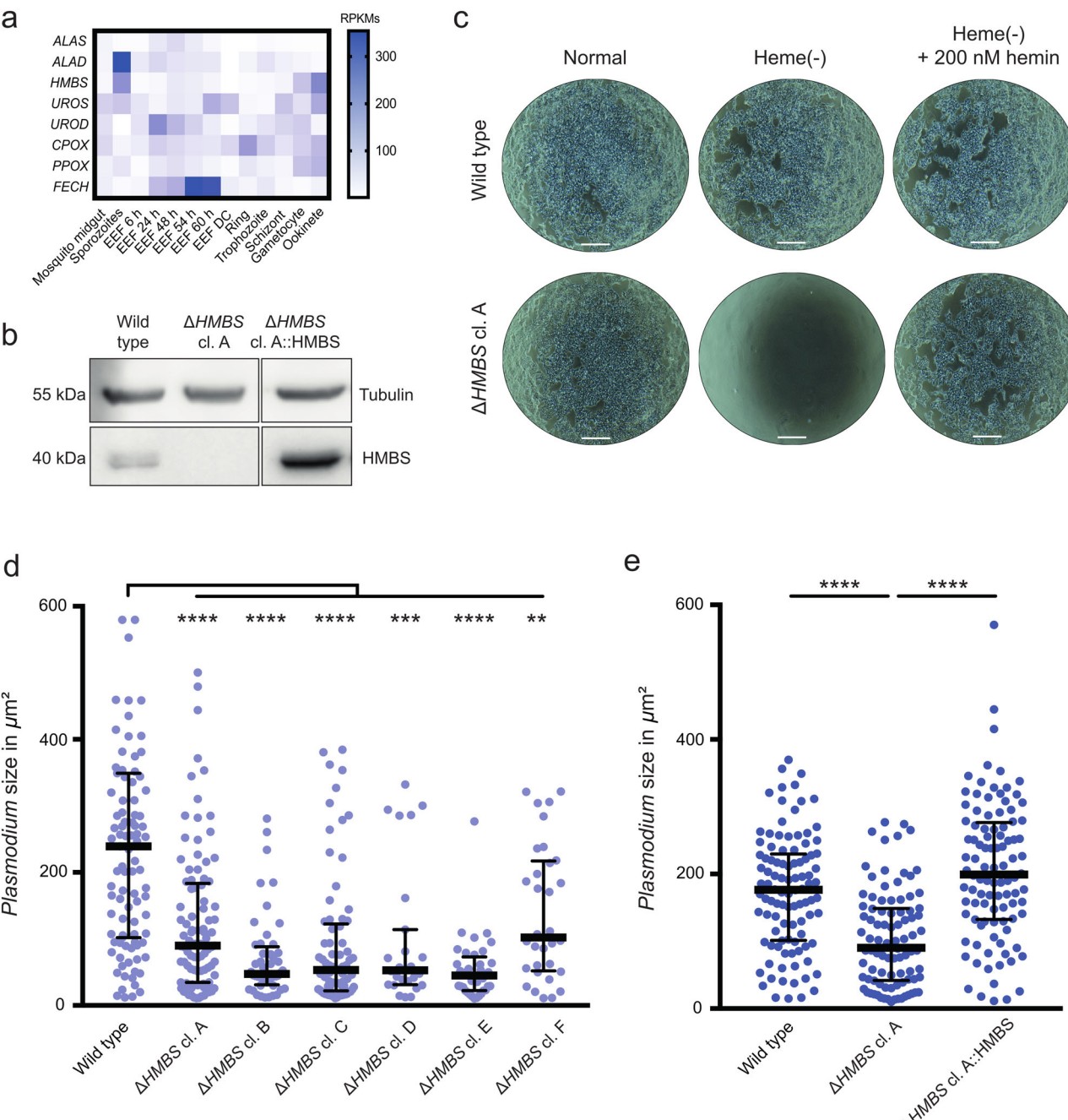

**Fig. 7 | Environmental heme from the host cell plays a key role in the development of *Plasmodium* exo-erythrocytic schizonts. a** Expression of *P. berghei* heme biosynthesis genes throughout the parasite's life cycle. Most parasite genes are expressed at low levels during the liver stage (EEF), except for *FECH*. EEF: exo-erythrocytic form; DC: detached mammalian cell containing merosomes. **b** Western blot of whole cell lysates from HAP1 wild type, Δ*HMBS* clone (cl.) A, and a pool of Δ*HMBS* cl. A::HMBS complemented cells. HMBS and tubulin (as control) were detected with specific antibodies as described in *Methods*. Scale bar = 60 μm. This experiment has been repeated multiple times (>3) with similar results. **c** Phase contrast images of HAP1 wild type and HAP1 Δ*HMBS* cl. A after 48 h incubation in normal 10% FCS-IMD media, heme-depleted 10% FCS-IMDM, and the latter with hemin supplementation. **d** *P. berghei* exo-erythrocytic schizont size at 48 h pi in HAP1 WT and six HAP1 Δ*HMBS* clones A - F. Cells were infected with mCherry-expressing *P. berghei* sporozoites and size was determined using automated imaging. The graph shows sizes of individual parasites ($n = 27–99$) with median and interquartile range. **e** Parasite size at 48 hpi in HAP1 WT, Δ*HMBS* cl. A, and Δ*HMBS* cl. A::HMBS complementation. Infection and size measurements were done as described in (**d**). The graph shows sizes of individual parasites ($n = 92$) and includes means with SD and *p* values of one-way ANOVA analysis with Dunnett's multiple comparison test.

## Liver-specific model for *Plasmodium falciparum*

The liver-iPfa model was reconstructed from the published iPfa genome-scale model for *P. falciparum*[26] by blocking the reactions that transport in and out of the parasite compounds that are not present in the liver model. By mapping the extra-parasitic metabolites from iPfa to the cytosolic metabolites from the hepatocyte model, we found that from the 241 metabolites that the parasite can transport in the iPfa model, 165 (Supplementary data 1) are present in the cytosol of the liver model. We then blocked the transport reactions for the 76 metabolites that are not found in the hepatocyte model and performed flux variability analysis to remove the reactions that could not carry flux in iPfa, resulting in a

liver-iPfa model of 737 metabolites, 889 reactions, 210 genes, and a growth of 0.073 h⁻¹ (doubling time of 9.48 h).

## Minimal nutritional requirements for liver-stage *Plasmodium falciparum*

In order to investigate the minimal set of nutrients that the liver-stage *P. falciparum* needs to sustain growth, we applied the in silico minimal medium (iMM) method[30] to the liver-iPfa model. The iMM method identifies the minimal set of nutrients that are required to satisfy a specific growth rate. In this case, we identified that parasites need a minimum of 31 nutrients to grow. Given the flexibility of the intracellular metabolism to synthesize the required precursors for biomass, there exist alternative sets of nutrients that could serve to achieve the same growth rate. In this case, using iMM we found 1792 alternative sets of 31 nutrients, leading to a total of 47 unique nutrients across sets. By classifying the presence of these nutrients in the corresponding 1792 sets, we identified 23 nutrients that appear in all sets, which we call constitutive, and 24 nutrients that are not appearing systematically in all sets because they can substitute each other, as they contribute with the same backbone moieties (Fig. S1a and Supplementary data 1).

In this scenario, when considering a liver-stage *P. falciparum* that takes up the minimal number of metabolites from its host cell, we are simulating a partial prototrophic parasite. Such a parasite can synthesize the maximum number of biomass precursors and will only scavenge from the host those that it cannot synthesize itself.

## The parasitosome as a representation of the metabolic state of *Plasmodium falciparum*

The parasitosome is a unidirectional metabolic reaction defined by the following steps:

1. Set the extracellular composition in the parasite model. We considered two cases: (i) parasites in an auxotrophic mode, which take up the available nutrients from the host cytosol, (ii) parasites in a partial prototrophic mode, which take up the minimum amount of nutrients from the host and use their intracellular metabolism to synthesize the rest of the compounds required for growth.
2. Use the lumpGEM method[51] in the liver-iPfa model to identify the set of intracellular reactions that must be active to simulate parasite growth. This subnetwork is known as minimal network (MiN).
3. Then, lumpGEM generates a lumped reaction by collapsing the subnetwork into one reaction (see original publication for more details). In the resulting reaction, substrates represent the nutrients that the parasite takes up from the host's cytosol, and products represent the by-products that the parasite must secrete into the host's cytosol.

It is important to note that the subnetwork generated by lump-GEM contains the details of the metabolic reactions active for the corresponding metabolic configuration as well as the biomass reaction of the parasite. As a result, one of the metabolites in the parasitosome reaction is the parasite biomass. Therefore, the optimized flux through the parasitosome in the integrated model represents the overall growth of the parasite by uptake and secretion of compounds from and to the host cytosol. Note that the parasitosome can support growth up to the maximum value but it can also capture suboptimal growth.

The directionality of each parasitosome, as well as the directionality of the reactions composing the underlying subnetwork are determined by lumpGEM. The algorithm finds the optimal configuration of fluxes that maximize biomass production in the parasite model, taking into account the directionality and thermodynamic feasibility of the reactions as defined in the curated GEM.

Note also that the alternative biochemistry of the parasite leads to alternative metabolic configurations (subnetworks) and thus to alternative parasitosomes.

In the case that we simulate an auxotrophic parasite, a minimum of 248 reactions are required and there exist 151 alternative MiNs (Fig. S1c). By lumping the MiNs into individual reactions, we generated 151 parasitosomes that use a total of 75 nutrients and secrete a total of 27 products (Fig. S1b). In the case where the parasite is in a partial prototrophic mode, the size of the MiNs is 307 reactions and there exist 8 alternatives (Fig. S1c), resulting in 8 alternative parasitosomes that take up a total of 36 nutrients and secrete 36 products (Fig. S1b).

## Reconstruction of a hepatocyte-parasite integrated metabolic model

The generated parasitosome reactions were integrated into the hepatocyte metabolic model. We then evaluated whether the bounds of the transport reactions in the hepatocyte model needed to be modified in order to sustain the growth of both the parasite and the hepatocyte. To do this, we formulated the following MILP:

$$objective\ function\quad \min \sum_{t=1}^{tpt\_rxn} \delta_{L_t} + \delta_{U_t}$$

subject to

$$FBA\ constraints$$
$$TFA\ constraints$$
$$fix\ biomass\quad v_{biomass_{host}}^F \geq \rho \cdot \mu_{host}$$
$$v_{biomass_{parasite}}^F \geq \rho \cdot \mu_{parasite}$$
$$relax\ bounds\quad v_{L_t} - \delta_{L_t} \leq v_t \leq v_{U_t} + \delta_{U_t} \quad \forall t \in tpt\_rxn$$

where *tpt_rxn* is the set of transport reactions in the network, *v* are the net fluxes for all the reactions and, $v_L$ and $v_U$ are the lower and upper bound, respectively, for all the reactions in the network. $\delta_{L_t}$ and $\delta_{U_t}$ are the relaxations of the lower and upper bound respectively. $\mu_{host}$ and $\mu_{parasite}$ are the theoretical maximum biomass for host and parasite, respectively and, $\rho$ is a percentage (0–100%).

As a result, the integrated model contains 3906 metabolites, 6377 reactions, and 1927 genes.

## Gene essentiality analysis in the integrated hepatocyte-parasite model

We performed gene essentiality analysis[52] in the healthy hepatocyte model and in the integrated hepatocyte-parasite model. In this analysis, hepatocyte genes are knocked out individually in the network. When a gene encoding for an enzyme is knocked out, the reaction rule is evaluated and if its rate is set to zero we assess whether the corresponding KO influences biomass production.

For the integrated hepatocyte-parasite model, we knocked out the hepatocyte genes and we evaluated the impact on biomass production for the hepatocyte and for each individual parasitosome. We then selected the genes that were essential for parasite growth but not for hepatocyte biomass production, nor for the healthy hepatocyte.

## Model reconstructions quality evaluation

The reconstructed models for hepatocyte and host-parasite in the different conditions were assessed to be compliant with community standards. The models passed the metabolic tasks defined for human metabolism[53] (Fig. S10a) and were evaluated using MEMOTE[54] (Fig. S10b).

## Cell culture

*T. annulata*-infected macrophages (TaC12; RRID:CVCL_2G84) were maintained in L15 medium (Gibco, Thermo Fisher Scientific, Waltham MA, USA), while uninfected bovine macrophages (BoMac)[55] and human embryonic kidney cells (HEK293FT; RRID:CVCL_6911) were grown in DMEM medium (Gibco). HAP1 cells (RRID:CVCL_Y019; gift from T. Brummelkamp, NKI, Amsterdam, The Netherlands), a human

near-haploid cell line derived from chronic myelogenous leukemia KBM-7 cells, were maintained in IMDM medium (Gibco). All media were supplemented with 10% heat-inactivated fetal calf serum (Bio-Concept, Allschwill, Basel, Switzerland), 2 mM L-glutamine (Bio-Concept), 10 U/mL penicillin-streptomycin (BioConcept). L15 and DMEM media were also supplemented with 10 mM HEPES pH 7.2 (Merck). All cell lines were grown at 37 °C in an atmosphere containing 5% $CO_2$, except for TaC12, which were cultured at 0% $CO_2$.

## Plasmids
All plasmids used in this study are listed in the Table S1.

## Human and bovine sgRNAs
All sgRNAs used in this study are listed in the Table S2.

## Primers
All primers were synthesized by Microsynth AG (Baldach, Switzerland). Primers for plasmids and TIDE PCRs are listed in the Table S3.

## Lentivirus production
The following plasmids were used for lentivirus production: psPAX2 packaging plasmid (Addgene, #12260), pMD2-G VSV-G envelope plasmid (Addgene, #12259), Blast-Cas9 plasmid, pKLV2-U6gRNA5(gGFP)-PGKBFP2AGFP-W reporter plasmid (Addgene, #67980), and custom pooled CRISPR libraries. Lentiviruses were produced using the following protocol. First, ~70% confluent HEK293FT cells in a 15 cm plate dish were transfected with transfer, packaging, and envelope vectors in a 4:2:1 ratio. 125 µl of 2.5 M calcium chloride was added to the plasmid mixture, and the final volume was made up to 1125 µl using 1 mM Tris, 0.1 mM EDTA, pH 8. The mixture was incubated at room temperature for 5 min. Then 1250 µl of 2 × HBS (50 mM Hepes, 280 mM NaCl) was added dropwise to the final plasmid solution while vortexing at full speed for 1 min. The mixture was immediately added dropwise to HEK293FT cells. Approximately 16 h post-transfection, the culture medium was replaced with fresh medium. The following day, ~30 h after media replacement, cell supernatants were collected and filtered through a 0.45 µM filter to remove cell debris. Virus-containing supernatants were ultracentrifuged for 2 h at 20,000 × g at 4 °C (Hitachi CP100NX ultracentrifuge) and the pellet was aliquoted in 1 × PBS (~200 µl for viruses produced from one 15 cm plate dish). Concentrated virus was aliquoted and stored at −80 °C. Viruses were harvested every 24 h (by adding fresh medium to the HEK293FT cells) for 3–4 days. The viral titer of concentrated viruses was calculated using the qPCR Lentivirus Titration Kit from ABM (Applied Biological Materials, Vancouver, Canada) according to the manufacturer's instructions.

## Generation of Cas9-expressing cell lines
To achieve constitutive Cas9 expression, TaC12 and BoMac cell lines were infected with lentiviruses containing the Cas9-Flag-Blast construct at a multiplicity of infection of 10. Briefly, 100,000–500,000 cells were centrifuged at 100 × g for 5 min. The supernatant was removed, and the cell pellet was resuspended in 1 ml medium containing 8 µg/ml of polybrene. Aliquots of lentivirus stored at −80 °C were thawed on ice and the appropriate volume of virions was added to the cells. Virus-cell solutions were incubated for 20 min at room temperature, followed by a 30 min spin-infection performed at 30 °C in a laboratory centrifuge (100 × g). After centrifugation, the pellet was resuspended in the same virus-containing medium and cells were seeded into 6-well plates. After 24 h of incubation, the supernatant was replaced with fresh medium containing blasticidin (Sigma-Aldrich, St. Louis MO, USA) to select transduced cells (10 µg/ml for TaC12, 5 µg/ml for BoMac). For each cell line, two controls were included in the experiment: positive control (non-transduced cells in polybrene containing medium) and negative control (non-transduced cells in blasticidin containing medium). When the negative controls reached 100% cell death, transduced blasticidin-

resistant cells were collected for validation of Cas9 expression by Western blotting. A fluorescence-based reporter assay was performed to specifically select cells expressing active Cas9 (Fig. S11). Briefly, Cas9-expressing cells were transduced with a lentiviral vector consisting of blue fluorescent protein (BFP) and green fluorescent protein (GFP) expression cassettes as indicated, together with a sgRNA targeting GFP. Cells were infected at an MOI of 10 in the presence of 8 µg/ml of polybrene for 24 h and seeded into T25 flasks. The culture medium was refreshed the following day. Between five and seven days after transduction, cells were sorted by FACS for BFP + GFP- signal, indicating expression of active Cas9 protein. For sorting, cells were resuspended in 1% FCS/PBS at a concentration of 10 mio cells/ml. BFP + GFP- cells were collected in a Falcon tube containing 100% FCS, centrifuged, and rapidly transferred to T25 flasks containing normal culture medium. Sorting was repeated twice for enrichment, using the MoFlo ASTRIOS EQ cell sorter (Beckman Coulter).

## CRISPR/Cas9 dropout screens
Genome-wide CRISPR/Cas9 dropout screens were performed in Cas9-expressing TaC12 cells using a two-vector system, as well as small-scale screens in TaC12 and BoMac cells. For the genome-scale screen we generated a CRISPR sgRNA containing 84,155 sgRNAs targeting 21,039 protein-coding genes, with 4 sgRNAs designed per each gene. For the small-scale screen we generated a CRISPR sgRNA library containing 16,596 sgRNAs targeting 1661 protein-coding genes, with 10 sgRNAs designed per each gene. Five hundred non-targeting and 500 intergenic sgRNAs were added to each CRISPR library as controls. The bovine libraries were cloned into lentiGuide-Puro plasmid (Addgene, #52963) and viruses were produced in HEK293FT cells. Screens were performed at a coverage of 500 and in biological triplicates. A starting population of at least 140 mio cells (genome-wide screen) or 60 mio cells (small-scale screen) was infected with the bovine lentiviral library in normal growth medium in presence of polybrene (8 µg/ml) for 24 h. The next day, the culture medium was replaced and fresh medium containing puromycin (2 µg/ml for TaC12; 1 µg/ml for BoMac) was added to the cells. In both genome-wide and small-scale dropout screens, a positive control (untransduced cells) and a negative control (untransduced cells treated with puromycin) were included for each replicate of each CRISPR screen in all bovine cell lines. After 3 days of puromycin selection the negative control reached complete cell death, while successfully transduced cells were >43 mio cells (e.g., 85,155 sgRNAs × 500) or >9 mio cells (e.g., 17,596 sgRNAs × 500). The lentivirus infection was optimized to ensure a transduction efficiency of ~30%, reducing the chance that a single cell expresses multiple sgRNAs. Puromycin-resistant cells were maintained in culture under drug selection for at least 11 passages and harvested at the first and last passages for genomic DNA (gDNA) extraction. gDNA was extracted from "START" and "END" point samples and high-throughput Illumina sequencing was performed to quantify the abundance of each sgRNA by PCR amplification of the sgRNA cassette. The resulting reads were deconvoluted using the PoolQ tool (https://github.com/broadinstitute/poolq) to generate a matrix of read counts, and transformed into log-norm files. Log2 fold change (LFC) values for each guide were then generated by subtracting "END" point from "START" point values. CRISPR screen data was analyzed using custom made chip files (Supplementary data 5, 6).

## Generation of HAP1 knockout cell line
For human *HMBS* knockout, we chose a sgRNA targeting an early coding region of the *HMBS* gene (5′-GCAGCTTGCTCGCATACAGA-3′) and cloned this guide into the pCRISPR_v3.1 plasmid. Low-passage HAP1 cells (<p5) were seeded at a concentration of 50,000 cells/well in 24 well plate. The following day, 600 ng of pCRISPR_v3.1+sgRNA (*HMBS*) plasmid was transfected using Lipofectamine3000 according to manufacturer's instructions. Twenty-four hours post transfection, the culture medium was refreshed and 0.6 µg/ml puromycin was

 

added to the transfected cells as well as in the negative control. Genomic DNA was extracted and the sgRNA (*HMBS*) target locus was PCR amplified using Q5 High-Fidelity DNA polymerase (New England Biolabs, Ipswich, Massachusetts) (forward primer: CAGAGGGT-TAGTTCCTAGTA, reverse primer: AGAGAGTGCAGTATCAAGAA). The resulting amplicon was purified using the Gel & PCR clean-up kit (Macherey-Nagel, Düren, Germany) and sequenced by Sanger sequencing. The online tool Tracking of Indels by Decomposition (TIDE) was used to interpret the sequencing results (http://tide.nki.nl). Limiting dilutions of the polyclonal HAP1 Δ*HMBS* pool were performed in 96 well plate, and six isolated clones were selected. Successful knockout of the *HMBS* gene was verified at genome level (identification of frameshift mutations at the target locus) and at the protein expression level (Western blotting).

### Re-expression of HMBS in HAP1 Δ*HMBS* clone A

The housekeeping isoform of HMBS transcript (ENST00000652429.1, HMBS_232) was amplified from complementary DNA synthesized by reverse transcriptase starting from RNA of HAP1 WT cells (GoScript™ Reverse Transcriptase, Promega, Madison WI, USA). HMBS_232 was cloned into pCMV-puroR lentiviral plasmid using the primers listed in Table S3 and NEBuilder HiFi DNA Assembly Master Mix (New England BioLabs). Lentiviruses were produced as already mentioned. HAP1 Δ*HMBS* cells were transduced for 24 h with virus-containing supernatants harvested from HEK293FT dishes, and cells were selected for 4 days with 0.6 μg/ml puromycin. Untransduced HAP1 Δ*HMBS* cells were used as a negative control and reached 100% cell death following 4 days of puromycin selection. Puromycin-resistant cells were subjected to limiting dilutions, and several single clones were analyzed via Western blotting for HMBS expression.

### Western blotting

Whole cell lysates were prepared by lysing 5 mio cells in 500 μl lysis buffer 50 mM Tris-HCl, 150 mM NaCl, 1 mM EDTA, 1% NP-40, 0.25% sodium deoxycholate, 1x cOmplete™ protease inhibitor cocktail (Roche, Basel, Switzerland). The samples were boiled for 5 min at 90 °C, and then cooled on ice. Whole cell lysates were then loaded onto 10% sodium dodecyl sulphate (SDS) polyacrylamide gels. Gel runs lasted 45–50 min at 25 mA in electrophoresis buffer (2.5 mM Tris, 19.2 mM glycine, 0.01% SDS), followed by protein transfer to a nitrocellulose membrane for 90 min at 400 mA in transfer buffer (20 mM Tris-HCl, 15 mM glycine, 20% (v/v) methanol). The membrane was incubated for several hours in blocking solution 5% milk powder in TBST (10 mM Tris, 150 mM NaCl, 0.05% Tween). Primary antibodies were diluted in blocking solution and incubated with the membrane overnight at 4 °C. Primary antibodies used in this study are: rabbit anti-HMBS (Sigma HPA050659, dilution 1:1000), mouse anti-tubulin (Sigma T9026, dilution 1:5000), mouse anti-Flag M2 (Sigma F3165, dilution 1:2000). After three washes with TBS (10 mM Tris, 150 mM NaCl), the membrane was incubated with secondary antibodies diluted in blocking solution for 45 min at room temperature. Secondary antibodies used in this study are: goat anti-rabbit HRP (Cell Signaling 7074 S, dilution 1:2000), rabbit anti-mouse HRP (Dako P0260, dilution 1:2000), donkey anti-rabbit (Licor 926-32213, dilution 1:10.000), goat anti-mouse (Licor 925-68070, dilution 1:10.000). After three further washes with TBS, the Western blot was then imaged using the C600 imaging system (Azure Biosystems).

### FCS heme depletion

Heme-depleted FCS was prepared by treating FCS with 20 mM ascorbic acid for 16 h at 37 °C, followed by 24 h dialysis against PBS. Heme depletion was verified by measuring optical absorbance at 405 nm[56].

### Cell viability assay

Cell viability assays were performed in 96 well plates. Briefly, 5000 TaC12 or BoMac cells were seeded into the 96 wells and cells were allowed to grow for 72 h under normal culture conditions in the presence of inhibitors or solvent alone. Cell viability was measured using the resazurin assay.

### Generation of BoMac and TaC12 knockout pools

To perform single CRISPR/Cas9 knockout of bovine candidate genes (*HMBS*, *GSS*, *SRM*, *VIM*) in TaC12 and BoMac cells, we first generated lentiviruses as described above ("Lentivirus production") using pCMV-puroR as vector plasmid (Table S1). This lentiviral plasmid contains Cas9 gene, puromycin resistance marker and sgRNA scaffold. Therefore, we generated four different pCMV-puroR plasmids, each one with a different bovine sgRNA cloned inside (Table S2) using NEBuilder HiFi DNA Assembly Master Mix (New England BioLabs). Virus-containing supernatants harvested from HEK293FT dishes were collected in 5 ml aliquots and stored at −80 °C. BoMac and TaC12 cells were seeded into T25cm flasks at a concentration of 50,000–100,000 cells/flask. The next day, the cells were transduced with 5 ml of lentiviruses for 48 h, with a fresh aliquot of lentiviruses added after 24 h. The culture medium was then replaced with fresh medium containing puromycin (2 μg/ml for TaC12; 1 μg/ml for BoMac). The negative control died after 3 days of drug selection, and successfully transduced cells were maintained in culture under puromycin selection until they reached confluence. After extraction of genomic DNA from the puromycin-resistant pools, the sgRNA target locus was PCR amplified using the primers listed in Table S3. Sanger sequencing results of the PCR amplified amplicons were analysed using the TIDE online tool.

### Mouse experiments and mosquito maintenance

Mouse studies were approved by the Commission for Animal Experimentation of the Cantonal Veterinary Office of Bern and conducted in compliance with the Swiss Animal Welfare legislation and under license BE118/22. Mice (BALB/cAnNRj) were purchased from Janvier-Labs (France) or bred in house and used at 8–12 weeks of age. They were kept in individually ventilated cages (Tecniplast, Italy) at 22 °C, 55% rH and 12 h/12 h dark/light cycle. Female mice were infected by i.p. injection of blood stabilates of mCherry expressing *Plasmodium berghei* parasites (RMgm-928, 1804cl1). At approximately 4% parasitemia, blood was collected by cardiac puncture and passaged by i.v. injection into male mice (feed mice). After 3 days, the parasitemia and gametocytemia were monitored by flow cytometry (NovoCyte, Agilent) and thin blood smear. At >0.4% gametocytes, the mice were anaesthetized (ketamin/xylazine) and made available to approximately 150 female *Anopheles stephensi* mosquitoes to take a blood meal for 30 min. The mosquitoes were maintained at 20.5 °C and above 80% relative humidity. Sporozoites were isolated from infected salivary glands on days 18–26 after the infective blood meal.

### *Plasmodium berghei* infection

HAP1 cells were cultured in IMDM (Bioconcept, containing 10% FCS, 4 mM L-glutamine, 100 U penicillin, 100 μg/ml streptomycin) at 37 °C, 5% $CO_2$. To infect confluent cultures, 70,000 cells were seeded in 96 well plates. The next day, the cultures were infected with 20,000 freshly isolated sporozoites. To allow development over 48 h, cultures were expanded by passaging with accutase (Innovative Cell Technologies) at 2 hpi. Cells were reseeded into 96 well plates (Greiner μClear, 655090). Each infected well was divided into 12 wells. At 48 hpi, cells were imaged with a Nikon Ti2 inverted fluorescence microscope Plan Apo λD 10× (NA 0.45) objective, Spectra X light engine (555 nm line, Lumencor), acquired with a Kinetix22 camera (Photometrix), Pinkel quad and Sedat quad filter sets (MXR00244 and MXR00254, Semrock). Parasite sizes were segmented and analyzed using the General Analysis Module of NIS-Elements 5.42.02. Graphs were generated with Prism 9 (GraphPad) and indicate medians with interquartile range. One-way ANOVA with Dunnett's multiple comparison was used as statistical evaluation, adjusted *p* values are presented directly in the graph.

## Software for data analyses and computation

The metabolic modeling computational analyses were performed using Matlab (MathWorks, v2021b) and CPLEX (IBM, v12.10). Data analyses and associated graphs were generated with Rstudio, Microsoft Excel, FlowJo, and Prism 9 (GraphPad).

## Reporting summary

Further information on research design is available in the Nature Portfolio Reporting Summary linked to this article.

## Data availability

CRISPR screen data for this study have been deposited in the European Nucleotide Archive (ENA) at EMBL-EBI under accession number PRJEB73635 and accession numbers for the RNA-seq data for uninfected macrophages (ERS16273013, ERS16273014, ERS16273015) and TaC12 cells (ERS18408949, ERS18408950, ERS18408951) (https://www.ebi.ac.uk/ena/). Chip files for the genome wide bovine library (CP1273, 85155 distinct barcodes) and the sublibrary (CP1504, 17596 distinct barcodes) have been uploaded as supplementary files. Source data are provided with this paper.

## Code availability

The code to generate the results for this paper is available under the APACHE 2.0 license at https://github.com/EPFL-LCSB/host_parasite_interactions. The code for TFA is available at https://github.com/EPFL-LCSB/mattfa. The code to identify minimal nutrients (iMM) is available at https://github.com/EPFL-LCSB/redhuman.

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

## Acknowledgements

We would like to express our gratitude to Francis Brühlmann for providing the IFA image of *Theileria annulata* used in Fig. 1. We thank the Flow Cytometry and Cell Sorting Facility (FCCS) of the Department for BioMedical Research (DBMR) of the University of Bern, Switzerland. Financial support came from the University of Bern (UniBe ID grant to S.R. and V.He.) and the Swiss National Science Foundation (198543 to S.R., V.He., V.Ha., 189127 to S.R., 173972 to P.O.). Figures were created with the help of BioRender.

## Author contributions

Conceptualization, K.W., S.R., P.O.; methodology, J.G.D., K.W., V.Hr., V.Ha., S.R., P.O.; investigation, M.Mr., M.Ms., K.W., R.C., A.N., D.J., J.Z., M.B.; formal analysis, M.Mr., M.Ms., K.W., R.C., A.N., M.G.F., P.O.; writing – original draft, M.Mr., M.Ms.; writing – review & editing, K.W., R.C., J.G.D., A.N., V.Hr., V.Ha., S.R., P.O.; funding acquisition, V.Ha., V.Hr., S.R., P.O.; resources, V.Ha., V.Hr., S.R., P.O.; supervision, V.Ha., V.Hr., S.R., P.O.

## Funding

## Competing interests

The authors declare no competing interests.
