## [Transparent Peer Review file · Nature Communications]

Host Cell CRISPR Genomics and Modelling Reveal Shared Metabolic Vulnerabilities in the Intracellular Development of *Plasmodium falciparum* and Related Hemoparasites

Corresponding Author: Professor Philipp Olias

Version 1:

Reviewer comments:

Reviewer #1

(Remarks to the Author)

In their manuscript, "host cell CRISPR genomics and cheminformatics reveal shared metabolic vulnerabilities in the intracellular development of *Plasmodium falciparum* and related hemoparasites, Maurizio and colleagues take an innovative and interesting approach to identifying common host regulators between *Plasmodium* liver stages and *Theileria* white blood cell infections. They use a metabolic modeling approach, taken from human hepatocyte data, then map known information about *Plasmodium* metabolism on top of this model. From this, they infer metabolic requirements of the *Plasmodium* liver stage, and compare this to hits in a whole genome CRISPR/Cas9 screen for host regulators of *Theileria* infection. This approach is very innovative and seeks to draw parallels between these related, but still reasonably distant parasites. However, many major leaps between experimental systems without validation remain, and as a result, many of the conclusions appear somewhat speculative.

Major comments:

1. A huge number of practical differences exist between the two parasite they study. For example, different host species and cell type, quite evolutionarily different genomes, and the presence (*plasmodium*) and absence (*theileria*) of a parasitophorous vacuole membrane, at the interface between parasite and host.
2. It seems that the jump between *Plasmodium* in hepatocytes and *Theileria* in macrophages is a huge one. Could a similar approach be used to derive the metabolism requirements of *Theileria* as a proof-of-concept in white blood cells, and then go back and directly compare to their whole genome CRISPR data? As a second step, could white blood cells and hepatocyte be more directly compared to hepatocytes? As the manuscript flows, it is a bit of a leap of faith for the reader to see the parallels between the two experimental systems.
3. One of the exciting parts of this manuscript is the whole genome CRISPR/cas9 that identifies host regulators of *Theileria*, but the authors present this only as a motivation to better understand *P. falciparum*. *Theileria*, itself, is an important parasite and I would like to see these hits validated further in the *Theileria* / bovine macrophage system.
4. Final validation experiments in Fig. 6 are performed in *P. berghei* – although the hope is to identify host regulators of *P. falciparum*. Rodent-infectious *Plasmodium* species serve a very important role in modeling *P. falciparum*, but this is somewhat confusing as multiple previous screens have identified host regulators of *P. berghei* and *P. yoelii* via siRNA and CRISPR/Cas9 approaches.
5. The authors cite (refs 11-13 multiple other whole-genome CRISPR screens that have identified host regulators of related parasites. Authors should compare their screen (Fig. 3) as well as their metabolic model (Figs 1-2) to these data to evaluate overlap. This is particularly relevant for the *Plasmodium* screens since they are done in the system that the authors seek to better understand.

Minor comments:

6. In many places throughout the text, the language is not entirely clear what observations are predicted by models and what observations are experimentally obtained. This needs to be fixed throughout the manuscript so the reader can more clearly follow where each conclusion has come from.
7. On line 143, authors refer to hepatocyte growth. However, primary hepatocyte typically do not grow in vitro – do they mean survival here?

Reviewer #2

(Remarks to the Author)

In this fascinating work by Maurizio and colleagues, the authors use a combination of molecular biology-based genome-

wide screen and in silico modeling of complex biological systems to identify host genes essential for the intracellular multiplication and survival of two distinct taxa of single-celled eukaryotic parasites, Plasmodium and Theileria, both hemoparasites in the phylum Apicomplexa. For Plasmodium, the authors model the biochemical interactions between the parasite and the host hepatocyte using a combination of whole cell, genome-wide metabolic network models available for human hepatocytes and for Plasmodium falciparum. Interactions are modeled based on metabolites and enzymes available in the host cytoplasm; the authors conduct simulations across two host cellular environments: one nutrient-rich and the other a nutrient-limited setting. They identify 209 and 110 genes to be essential for parasite growth, respectively, in the former and the latter settings, and 19 genes essential under both conditions. For Theileria, they use as model Theileria annulata-infected host macrophages, and conduct an CRIPR/Cas9 genome-wide KO screen to identify the host genes essential for Theileria growth, of which they find several hundred. Finally, the authors identify 24 and 7 host enzymes essential for both parasites, when Pf is grown respectively in nutrient-rich and -depleted cellular environments. In the latter case, enzymes belong to heme and purine biosynthesis pathways, which the authors go on to validate experimentally in an in vivo Plasmodium mouse model. This paper reflects a tremendous amount of very solid work. Its modeling predictions reveal metabolic preferences for Plasmodium growth under each environment and, interestingly, reveal the dependence on host heme biosynthesis during Plasmodium liver stages. A few suggestions that may improve the manuscript are listed below.

Minor Questions and Issues:

1. I find it curious that more genes are “essential” for parasite growth in a rich host cell environment. Is the parasite predicted to die/stop replicating if any of those is missing or can the parasite then switch to relying on its own metabolism? Ei, is the “nutrient-rich” environment actually toxic in some way if some of these putative “essential” host genes is missing, or are these genes considered essential just for the parasite to live on a ‘scavenger’ metabolic mode? This may be worth clarifying.
- 2.
3. L191. “screen was performed at a coverage of 500, with” – pls explain; I assume a ratio of 500 sgRNA per cell but currently too cryptic
4. Fig 2: “grey” bar in panel H is invisible (mark with darker grey)
5. Fig 3, panel C.
 - a. Pls explain what are the variables assessed in each replicate (I assume reads/locus, but this needs to be made clear).
 - b. Virtually no correlation between replicates. Why?
6. Fig 3, panel D. green and blue colors are difficult to distinguish. Change one of them?...
7. L206: “log-normalized counts confirmed...” : again, pls explain what the counts reflect
8. Fig 4, panels E-G:
 - a. resolution is poor (can’t see text in some axes or in the pathways);
 - b. add legend for red/green/blue/black bar graph; also, numbers within bar graph are not readable.
9. L302-329: Section on “Perturbation of environmental heme pools...” : is hemin a chemical homolog for heme or for one of its precursors? The reason I ask is that I wonder if it is possible that the parasite doesn’t grow in HMBS-deficient cells not because of the absence of heme per se but because the cells are damaged/deficient in a parallel pathway/molecule dependent on hydroxymethylbilane (or one of the other heme precursors). This question is resolved if hemin is replacing heme directly. Pls clarify.

Minor issues/typos:

10. L50: Humans are animals... Replace “in humans and animals.” with “in humans and other animals.”
11. L50-53: Since Plasmodium represents a whole group of species, it is not appropriate to refer to it as a “parasite”. Rephrase sentences to read: “Malaria-causing Plasmodium falciparum is the most impactful human parasite, with [...]. Other socio-economically important apicomplexans are the closely related livestock parasites in the genus Theileria, which kill more than 1 million (Mio) cattle each year and have ...”
12. L70: Spell out complete species names the first time they are mentioned (Plasmodium yoelii <- notice typo in manuscript); L89-90: Theileria annulata
13. L165: “show that SRM is” -> “show that spermidine synthase (SRM)” : I suggest spelling out gene name (especially since that is done in the rest of the sentence);
14. Be consistent throughout on italicizing genes but not their products (that seems in L165-167 as I assume that in L165 the authors are referring to the gene) but there are places where this breaks down (eg, L232-234).

Reviewer #3

(Remarks to the Author)

NCOMMS-23-52726A-Z

The manuscript entitled “Host Cell CRISPR Genomics and Chemoinformatics Reveal Shared Metabolic Vulnerabilities in the Intracellular Development of Plasmodium falciparum and Related Hemoparasites” by Maurizio et al. describes an experimental and computational framework to understand host-parasite interactions during the infection Plasmodium and Theileria. The manuscript has impressive modeling work as well as experimental screenings of Theileria. Despite the efforts of the authors to connect the modeling outcomes of both host-pathogen interactions the workflow seems to have several missing parts in the P. falciparum site. If authors change the focus to Theileria rather than Plasmodium the story will be more comprehensive or in the worst-case scenario separate the two stories. Modeling of host-P. falciparum can be only an additional analysis. This new arrangement will help the organization of the manuscript because currently there are several jumps back and forth between the two systems. Lastly, testing of models architecture and simulations was not possible because models were missing in the submission and the lack of repository information.

The introduction of the concept parasitome is interesting, however host-pathogen genome scale metabolic models have

been previously reconstructed (10.1038/msb.2010.68). Those models usually create a sub compartment that allows for exchange reactions among the host and parasite while enabling the flexibility to independently optimize for two or more biomass objective functions. Details about reaction reversibility and definition of the parasitome are missing or hard to find in the manuscript. It would be great if the authors add more details about this step.

The generation of the parasitome seems to have several limitations, for example the dependency on the fixed constraints (e.g. growth rate), which highly affects the feasibility of gene essentiality prediction. Another remarkable limitation is the optimization of a single biomass objective function. The lumpGEM method that authors previously published and implemented for this work enables the simulation of several physiological states, however algorithms such as OptCom are more broadly used. It would be interesting to make a comparison between the outcomes of both algorithms.

Minor comments

Some introductory sentences of the discussion will be highly beneficial in the introduction section. For example "Diverse heme acquisition strategies have been described among Apicomplexan parasites. While Plasmodium and Toxoplasma possess a complete and functional de novo heme biosynthetic pathway, parasites such as Babesia, Theileria and Cryptosporidium depend on acquiring heme from the host cell"

The manuscript is missing a figure to connect both stories describing the methods and results. For example Fig. 1-4 focuses on Plasmodium while only Fig. 4 with a totally different format focuses on Theileria data.

Figure 5 should be expanded to account for the results of Theileria.

Line 126. A clarification if the reduction of 165 to 47 nutrients is based on experimental evidence is missing.

Fig.2. Please clarify if any of the parasitomes and selected pathways was selected based on experimental data or only simulations.

Fig. 4D. Please use the models to perform this network analysis and move the current figure to supplementary materials. Insights about how models are going to be made publicly available is missing. Please describe the formats and repositories in which the models, code and workspaces will be shared.

Reviewer #4

(Remarks to the Author)

The goal of this study was to identify shared vulnerabilities for intracellular apicomplexan parasite development. The authors used a novel hepatocyte metabolic reconstruction to investigate dependency of Plasmodium falciparum on liver metabolism. This approach took advantage of tools previously developed by the authors including a parasite metabolic reconstruction (iPa) and a method to generate subnetworks (LumpGem) as well as a clever exploration of rich vs compromised conditions. The authors then assessed the effects of a CRISPR KO screen to explore the metabolic dependencies of a related apicomplexan parasite (Theileria) in its host macrophage. The top hits were validated using biochemical inhibitors. Together, the authors identify a core set of host metabolic genes required for parasite survival including purine and heme biosynthesis. However, an overall lack of rationale and discussion of chosen approaches limit the potential for broader appreciation, and thus significance; much more description is required to be useful for the community. Also, the major biological conclusions from these studies are modest given that it is well established that purine salvage is a major parasite weakness. The conclusions do contribute a viewpoint to the growing story about heme salvage requirements but do not finalize this story.

Major concerns:

1. There are numerous new models and concepts present in the manuscript that are not validated or are missing rationale or justification. Addressing these 'sticking points' are critical to ensure that the broader audience appreciates the choices that the authors have made during their approach. For example:

-The manuscript describes a novel GEM of hepatocyte metabolism but does not present any validation that this model is useful and accurate. Entire papers explore the creation and validation of models following standards of the field.

-A new concept in the paper is the idea of the "parasitosome", which are defined using a tool the authors developed previously (LumpGEM). This is the first time the tool has been applied to the parasite/intracellular context and the authors do not describe the rationale behind using the parasitosome concept in the context of the liver.

-Reconstructing the metabolism of an intracellular parasite is inherently complex and requires many assumptions. The manuscript lacks details that justify the chosen approach involving mapping "extraparasitic metabolites" to the liver cytosol. The attempt to explain the approach (line 552) is unclear while Fig 1 legend is somewhat better. Is there precedence in the literature that justify this route?

-While the comparison of rich vs compromised conditions is a useful comparison to explore the metabolic capabilities of the parasite, the manuscript is missing any physiologically justification of these conditions.

2. The manuscript is missing a discussion of the limitations of the bovine macrophage system. It is left to the reader to piece together that two different cell lines must be used as uninfected and infected systems (BoMAC and TaCl2- this information is not defined until the Methods section, see minor point below). Perhaps there is a reason behind this (i.e. cell line availability). However, comparison of these different cell lines leaves the door open that differences in essential genes are due to host cell lineage and not solely infection status. The manuscript would benefit from more details on this.

3. Again, the lack of rationale across the paper opens major questions for this reviewer. The computational approach using the integrated hepatocyte-Plasmodium model is distinct from the "Theileria essentialome" approach during the CRISPR screen, yet they are directly compared at the end.

-In the first half of the paper, the authors identify genes essential to parasite, yet dispensable to hepatocyte. This point is clearly described, but not justified (and convoluted across the paper- see Figure 5).

-Then in the second half of the paper, the authors take essential genes from one cell line (BoMacs) and subtract these from list of essential genes from a different cell line (TaC12) that is infected with Theileria. To this reviewer, this comparison yields genes that are essential for the macrophage when Theileria is around. The authors again leap from -genes essential in Theileria-infected macrophage- to "Theileria essentialome" (line 214), which is misleading and unjustified.
-Comparing essential genes for the parasite to essential genes for the macrophage is less worthwhile than comparing genes essential to host cells -hepatocyte to macrophage- or parasites -Plasmodium to Theileria-.

Minor concerns:

1. Line 82: Over the past several years, metabolic reconstructions have become an exciting way to explore parasite metabolism. Many references are missing here.
2. Line 126: The manuscript is missing justification for the two scenarios.
3. The reference to "case study I/II" in figures is not consistent with text (line 126, called "scenarios")
4. Line 134: should the figure reference be Fig S1C-D?
5. Line 191: rephrase to "500x coverage of the genome"
6. Line 210 "About 80% of candidate genes were validated as crucial for parasite survival (Fig. S4F)." -What candidate genes? From what cell line? The total 1500 tested? Be clearer here.
7. Line 233: "While enzymes that convert IMP to AMP are present (ADSS and ADSL), those deputed to synthesize GMP (IMPDH and GMPS) are absent". -need more clarity here: absent/present from pathways, or essentiality lists?
8. Line 238: need to define GSS and GCLC before reference in Results.
9. Line 266: FECH definition
10. Line 279: How do the authors justify 99 chosen enzymes? Stated earlier that 653 essential genes defined as "essentialome", 15% were metabolic- make this point clearer
11. Fig 4/S4, Panel A- choice of the two greens (sage and neon) are confusing and cannot be distinguished by this reviewer.
12. The manuscript would benefit from some additional biological details including pathways for heme and glutathione biosynthesis to better convey how these pathways are used in the parasites.

Author Rebuttal letter:

We thank the editor and the reviewers for their positive feedback and for their insightful and constructive comments how to further improve our manuscript. In our point-by-point response we have addressed all reviewers's comments. The responses to the reviewers's comments, proposed improvements, suggestions, and modifications are highlighted in blue below. In addition, we have highlighted the modifications made in the manuscript to incorporate the reviewers's suggestions.

REVIEWER 1

In their manuscript, host cell CRISPR genomics and cheminformatics reveal shared metabolic vulnerabilities in the intracellular development of Plasmodium falciparum and related hemoparasites, Maurizio and colleagues take an innovative and interesting approach to identifying common host regulators between Plasmodium liver stages and Theileria white blood cell infections. They use a metabolic modeling approach, taken from human hepatocyte data, then map known information about Plasmodium metabolism on top of this model. From this, they infer metabolic requirements of the Plasmodium liver stage, and compare this to hits in a whole genome CRISPR/Cas9 screen for host regulators of Theileria infection. This approach is very innovative and seeks to draw parallels between these related, but still reasonably distant parasites. However, many major leaps between experimental systems without validation remain, and as a result, many of the conclusions appear somewhat speculative.

We thank Reviewer 1 for the insightful review and overall positive feedback on our manuscript. We appreciate the recognition of our innovative approach to identifying common host regulators between Plasmodium falciparum and Theileria infections. We understand the concerns regarding potential jumps between experimental systems and acknowledge the importance of addressing these issues. In our rebuttal, we thoroughly address each specific concern and provide additional evidence to support our conclusions.

Major comments:

1. A huge number of practical differences exist between the two parasite they study. For example, different host species and cell type, quite evolutionarily different genomes, and the presence (Plasmodium) and absence (Theileria) of a parasitophorous vacuole membrane, at the interface between parasite and host.

We appreciate the reviewer's input on this matter. Reviewer 1 has rightly pointed out the disparity between Theileria and Plasmodium spp., two distinct apicomplexan parasites. While they differ significantly, our study does not aim to directly compare these related hemoparasites. Rather, our focus lies in elucidating a potential overlap in the metabolic necessities of two of the most destructive apicomplexan parasite diseases globally. For this purpose, we focus on their respective infection stages in nucleated host cells. To clarify this point, we have meticulously restructured Figure 1 in the revised manuscript and rewritten the abstract and introduction to further emphasise

the aim of our study (lines 33-34, 63-73):

The updated Figure 1A now illustrates the rationale behind our selection of *Plasmodium falciparum* and *Theileria annulata* for this study:

- (1) Phylogenetic closeness within the Apicomplexa phylum.
- (2) Significance as major pathogens, causing life-threatening malaria in humans and theileriosis in cattle.
- (3) Both parasites exhibit a schizont stage that develops within nucleated cells, a stage highly relevant for pathogenesis (*Theileria*) and the production of numerous new parasites within the host (*Theileria* and *Plasmodium*).

-2-

(4) Utilization of state-of-art methodologies, such as genome-scale modeling for schizont-infected host cells (*Plasmodium falciparum*) and in vitro CRISPR screening for schizont-infected host cells (*Theileria annulata*), to identify shared essential metabolic host genes for parasite development and survival.

Our multidisciplinary approach aims to uncover common vulnerabilities of apicomplexans, such as potential drug targets, that could enable the targeting of both parasitic diseases using the same compounds while preserving host integrity.

2. It seems that the jump between *Plasmodium* in hepatocytes and *Theileria* in macrophages is a huge one. Could a similar approach be used to derive the metabolism requirements of *Theileria* as a proof-of-concept in white blood cells, and then go back and directly compare to their whole genome CRISPR data? As a second step, could white blood cells and hepatocyte be more directly compared to hepatocytes? As the manuscript flows, it is a bit of a leap of faith for the reader to see the parallels between the two experimental systems.

We thank the reviewer for raising this concern. Indeed, the same methodology could be used to study the metabolic requirements of *Theileria* in bovine macrophages. However, to date, there are no bovine or *Theileria* genome-scale metabolic models (GEMs), which hinders the possibility of using the same approach (this has been clarified in lines 210-213) We agree with the reviewer that it would be excellent to perform such studies, but deriving and validating high quality bovine and *Theileria* GEMs would require significant research efforts that are beyond the scope of this manuscript. A major advantage of *Theileria* and the bovine host cells is the possibility to perform genome-scale CRISPR-Cas9 screens to experimentally validate the host cell essentialome. Such an unbiased, genome-wide functional experiment is currently impossible for *Plasmodium*. In contrast to *Theileria*, however, we have a strong basis for GEMs using *Plasmodium*-infected hepatocytes. We therefore believe that our computational modelling is an elegant approach to bridge both systems.

To address the second point, we performed additional experiments and investigated the gene expression of *Theileria*-infected versus non-infected macrophages and we compared the expression of metabolic genes between human white blood cells and hepatocytes. The following data analyses were added to the Supplement (Figure S4):

- (1) Deregulation of metabolic gene expression for uninfected vs. *Theileria*-infected macrophages.

-3-

By comparing the expression of metabolic genes in uninfected bovine macrophages and *Theileria*-infected macrophages, we observed that several metabolic pathways are deregulated upon infection. This highlights the relevance of studying metabolic pathways in infected cells to find new drug targets.

- (2) Metabolic gene expression of white blood cells and hepatocytes using the Human Protein Atlas (<https://www.proteinatlas.org/>).

Number of metabolic genes expressed in hepatocytes 1568

Number of metabolic genes expressed in macrophages 1618

Number of metabolic genes expressed in both cell types 1365

Number of metabolic reactions expressed in hepatocytes 4941

Number of metabolic reactions expressed in macrophages 4884

Number of metabolic reactions expressed in both cell types 4341

According to our analysis, more than 84% of the metabolic genes are conserved in both cell types, and both cells share 87% of the metabolic reactions. While there is obviously no perfect match between these different cell types, we believe that there is sufficient overlap to justify our unique approach to comparing these different host cell types.

3. One of the exciting parts of this manuscript is the whole genome CRISPR/Cas9 that identifies host regulators of *Theileria*, but the authors present this only as a motivation to better

-4-

understand *P. falciparum*. *Theileria*, itself, is an important parasite and I would like to see these hits validated further in the *Theileria* / bovine macrophage system.

We agree with the reviewer's recognition of the importance and impact of *Theileria*. The hits obtained from the genome-wide CRISPR screen were validated in a subsequent subscreen, and the results are available in the Supplementary Material (Fig. S4F and Table S3). In addition, three validated hits from the subscreen were confirmed at the individual gene level and validated by individual inhibitor assays, as shown in Figure 4. We agree that this CRISPR screen represents an excellent resource for *Theileria* research, and we are actively following up on some of the identified hits. As mentioned above, the aim of this particular study was to identify shared vulnerabilities for the two apicomplexans, and the use of modeling (for *P. falciparum*) and CRISPR screening (for *T. annulata*) represented the most advanced technologies to tackle the comparative genome-wide metabolic analysis of these two highly important parasites. The two different approaches worked independently and yielded a unique list of host genes - predictions in the case of *P. falciparum* and experimental CRISPR screening evidence for *T. annulata* - that are essential for intracellular survival of the parasites in nucleated host cells. For clarification, we have updated Figure 1 to indicate that the genome-wide CRISPR screen in *Theileria*-infected macrophages was not performed to validate the essentiality predictions made with the *P. falciparum* hepatocyte model. Further clarification has also been provided in the revised main text of the manuscript in lines 67-76: "We hypothesized that these auxotrophs, depending on their life cycle stage, rely on common host cell-derived metabolites for intracellular survival. Therefore, we explored the potential of targeting a single host metabolic pathway to block infection. Focusing on arguably the two most impactful genera globally, *Plasmodium* and *Theileria*, we investigated shared dependencies on key host metabolic proteins at the genomic scale (Fig. 1A). A common feature of both parasites is their initial expansion within nucleated host cells (the schizont phase) before massive expansion in red blood cells. This first expansion occurs within hepatocytes (*Plasmodium*) and white blood cells (*Theileria*), making it an ideal life cycle stage for comparative analysis."

4. Final validation experiments in Fig. 6 are performed in *P. berghei* although the hope is to identify host regulators of *P. falciparum*. Rodent-infectious *Plasmodium* species serve a very important role in modeling *P. falciparum*, but this is somewhat confusing as multiple previous screens have identified host regulators of *P. berghei* and *P. yoelii* via siRNA and CRISPR/Cas9 approaches.

The novelty of our work lies in generating a metabolic model of *P. falciparum*-infected human hepatocytes, yielding a list of host genes predicted to be essential for parasite growth in the human host. Currently, experimental work with *P. falciparum* sporozoites is highly restricted to very high-level biosecurity (BSL3) facilities, making it unfeasible to validate the hits experimentally with the human malaria parasite within our scope. Nonetheless, we considered it essential to investigate whether the host heme biosynthetic pathway could influence the development of the rodent malaria parasite (see also #5 below).

5. The authors cite (refs 11-13) multiple other whole-genome CRISPR screens that have identified host regulators of related parasites. Authors should compare their screen (Fig. 3) as well as their metabolic model (Figs 1-2) to these data to evaluate overlap. This is particularly relevant for the *Plasmodium* screens since they are done in the system that the authors seek to better understand.

We appreciate the reviewer's point regarding the importance of comparing the data generated by our approach with previously published screening data from related apicomplexan parasites to identify potential overlap. However, it is critical to consider the markedly different experimental setups of each study, particularly the distinct parasite life cycle stages, which in our opinion make direct comparisons not meaningful:

-5-

In the paper by Vijayan et al., 2022, the authors performed a genome-wide CRISPR KO screen in HepG2 cells transduced with a pooled library and drug-selected for several days. Subsequently, the cells were infected with drug-resistant GFP-expressing *P. yoelii* sporozoites and sorted based on GFP signal intensity at 24 hpi. Notably, this early time point does not capture the information about host metabolic pathways essential for *Plasmodium* schizont development. At 24 hpi, *Plasmodium* parasites are still in an early developmental stage, and most of the replication leading to schizont formation occurs later (approximately 48 - 60 hpi). In contrast, our CRISPR screen in *Theileria*-transformed cells had a different setup, with transduced cells cultured for 11 - 12 passages, allowing for multiple rounds of replication. In addition, *Plasmodium* essentiality predictions from the *Plasmodium* hepatocyte GEM only consider the metabolic requirements of the parasite.

In the work by Wu et al., 2020, a genome-wide CRISPR KO screen was performed in HFF cells, followed by infection with *T. gondii* tachyzoites for 10 days. Surviving cells were harvested, and genomic DNA was sequenced. Comparison with the *Theileria* essentialome and the *Plasmodium* predictions revealed a rather small overlap:

Overlap *P. falciparum* predictions
Overlap *Theileria* essentialome // *T. gondii* screen (Wu et al., 2020)
T. gondii screen (Wu et al., 2020) prototrophic auxotrophic
parasite parasite
ACTL6A AADAT AADAT
ATP6V1B2 AKR1A1 AKR1A1
CHD8 CPT1A CPT1A
FAM210A MTAP CYP4F12
GTF2H2 SDHC DECR2
HTATSF1 DHCR7
NDUFS7 GCAT
NEDD1 KL
PIM1 MTAP
POC1A SDHC
PSMB8 SLC4A1
RCC1
SDHC
SLC20A1
SRBD1

In the paper by Gibson et al., 2022, the authors conducted a genome-wide CRISPR KO screen in HCT8 cells, which were transduced with a pooled library and subjected to drug selection for several days. The cells were then infected with *C. parvum* sporozoites at a 90% kill dose. Surviving cells at 72 hpi were allowed to proliferate. This process was repeated for three rounds of infection and expansion. Genomic DNA was then extracted and sequenced. Comparison with the *Theileria* essentialome and the *Plasmodium* predictions identified only one host gene each:

Overlap *P. falciparum* predictions
Overlap *Theileria* essentialome // *C. parvum* screen (Gibson et al.,
C. parvum screen (Gibson et al., 2022)
2022) auxotrophic prototrophic
parasite parasite
TMEM30A SLC35B2
-6-

For these reasons, we decided not to include a comparative analysis with previously published data from these other parasites in the manuscript. However, in the discussion of our revised manuscript, we explain these differences in more detail and explain how our study complements the currently available data. On lines 391-397 we now write: "Interestingly, we identified very little overlap with previously published CRISPR-screens, such as the study by Vijayan and colleagues where they performed a genome wide CRISPR screen in *P. yoelii* infected cells. While their experiment was designed to identify host factors important for early stages of liver stages and development (24 hours post infection), our model focused on metabolic pathways essential for *Plasmodium* schizont development."

Minor comments:

6. In many places throughout the text, the language is not entirely clear what observations are predicted by models and what observations are experimentally obtained. This needs to be fixed throughout the manuscript so the reader can more clearly follow where each conclusion has come from.

We thank the reviewer for this comment, and we apologize for the lack of clarity. We have now clearly specified when the results are conclusions from model predictions and when they are conclusions from experimental observations (for example, see lines 168, 197, 248). We have also slightly changed the title of the manuscript to make this clearer.

7. On line 143, authors refer to hepatocyte growth. However, primary hepatocyte typically do not grow in vitro - do they mean survival here?

The authors thank the reviewer for pointing this out. We have corrected this in the main text, lines 164-166, which now reads: "The reconstructed models were then used to investigate which hepatocyte metabolic genes are essential for *P. falciparum* proliferation but dispensable for

REVIEWER 2

In this fascinating work by Maurizio and colleagues, the authors use a combination of molecular biology-based genome-wide screen and in silico modeling of complex biological systems to identify host genes essential for the intracellular multiplication and survival of two distinct taxa of single-celled eukaryotic parasites, Plasmodium and Theileria, both hemoparasites in the phylum Apicomplexa. For Plasmodium, the authors model the biochemical interactions between the parasite and the host hepatocyte using a combination of whole cell, genome-wide metabolic network models available for human hepatocytes and for Plasmodium falciparum. Interactions are modeled based on metabolites and enzymes available in the host cytoplasm; the authors conduct simulations across two host cellular environments: one nutrient-rich and the other a nutrient-limited setting. They identify 209 and 110 genes to be essential for parasite growth, respectively, in the former and the latter settings, and 19 genes essential under both conditions. For Theileria, they use as model Theileria annulata-infected host macrophages, and conduct an CRISPR/Cas9 genome-wide KO screen to identify the host genes essential for Theileria growth, of which they find several hundred. Finally, the authors identify 24 and 7 host enzymes essential for both parasites, when Pf is grown respectively in nutrient-rich and -depleted cellular environments. In the latter case, enzymes belong to heme and purine biosynthesis pathways, which the authors go on to validate experimentally in an in vivo Plasmodium mouse model. This

-7-

is paper reflects a tremendous amount of very solid work. Its modeling predictions reveal metabolic preferences for Plasmodium growth under each environment and, interestingly, reveal the dependence on host heme biosynthesis during Plasmodium liver stages. A few suggestions that may improve the manuscript are listed below.

We would like to thank the reviewer for the thorough review and appreciation of our work. We are pleased that Reviewer 2 found our approach intriguing and commendable. We have carefully considered the suggestions for improving the clarity and impact of our manuscript and addressed each suggestion point by point.

Minor comments:

1. I find it curious that more genes are "essential" for parasite growth in a rich host cell environment. Is the parasite predicted to die/stop replicating if any of those is missing or can the parasite then switch to relying on its own metabolism? Ei, is the "nutrient-rich" environment actually toxic in some way if some of these putative "essential" host genes is missing, or are these genes considered essential just for the parasite to live on a "scavenger" metabolic mode? This may be worth clarifying.

We thank the reviewer for this thoughtful question. The host genes that are essential across all parasitosomes are genes whose absence would cause the parasite to die (or stop replicating). On the other hand, host genes that are essential for some parasitosomes but not for others point to genes that are essential for the survival of the parasite under certain metabolic configurations, meaning that in their absence the parasite could potentially change its metabolism and adapt to the new environment. The increased gene essentiality (higher number of essential genes) in a "rich host cell environment" is attributed to the parasite living in a scavenger metabolic mode requiring the metabolic products of these genes. In this context, some genes are considered "conditionally" essential, meaning that in the absence of these genes, the parasite could adapt to produce the nutrients provided by the reactions catalyzed by these genes. Conversely, there are genes that are "consistently" essential, meaning that they are vital to the parasite even under minimal conditions, and the parasite cannot adapt if these host genes are knocked out. To clarify this, we have renamed the two conditions, changing the previous names of rich and compromised medium to the following: case study I = parasites operating in auxotrophic mode, case study II = parasites operating in a partial prototrophic mode.

We have added this to the main text in lines 137-142: "One must consider the parasite's ability to adapt its metabolism within the host cell to identify effective host gene targets for inhibiting parasite proliferation. For instance, *P. falciparum* demonstrates versatility by scavenging some nutrients from the host cytosol (as a auxotrophic parasite) or synthesizing them when unavailable (as a partial prototrophic parasite). This adaptability is crucial to understand when designing compounds that target host enzymes responsible for nutrient production."

And we have explicitly explained the higher essentiality in the case of auxotrophic parasites (previously called parasites growing in a rich extra-parasitic space) as follows in lines 178-181: "The higher essentiality observed in the case of auxotrophic parasites stems from their greater dependence on host metabolism compared to partial prototrophic parasites, which utilize their own metabolism for survival."

3. L191. âscreen was performed at a coverage of 500, withâ â pls explain; I assume a ratio of 500 sgRNA per cell but currently too cryptic

The text has been modified as follows in lines 225-226:

âThe genome-wide screen was performed at a coverage where each sgRNA is expressed in at least 500 cells and in three biological replicates.â

-8-

4. Fig 2: âgreyâ bar in panel H is invisible (mark with darker grey)

We have changed the figure accordingly.

5. Fig 3, panel C. (a) Pls explain what are the variables assessed in each replicate (I assume reads/locus, but this needs to be made clear).

Raw counts from each replicate were used as the data set for this graph. This information has been added to the legend of Figures 3C and 4B (lines 531-533, 549).

5. Fig 3, panel C. (b) Virtually no correlation between replicates. Why?

It is not correct that there is virtually no correlation between replicates. Since these are biological replicates (three independent transductions on different days using different aliquots of bovine CRISPR library lentiviruses), the observed variations are within the normal range of CRISPR screens. We think that our previous representation may have been misleading. The graph has now been modified to make it more readable, and the legends explaining the colours have been added (lines 532-533, 549-551)

6. Fig 3, panel D. green and blue colors are difficult to distinguish. Change one of them?...

We thank the reviewer for pointing this out. Colours have been changed to orange and green.

7. L206: âlog-normalized counts confirmedâ : again, pls explain what the counts reflect

We think that explaining these analytical details in the main text is disrupting the flow of the text.

The underlying analytic tools have been published and we have included the repository (line 839).

We therefore kindly refer the reviewer to the Material & Methods section (CRISPR/Cas9 dropout screens). In brief, the reads obtained from the Illumina sequencing were deconvoluted using the PoolQ tool to generate a matrix of read counts. These counts reflect the number of times a sgRNA was detected in a population, and thus constitute a measure of how many surviving cells harbour a sgRNA. These read counts are then normalized using a formula that takes into account the total number of reads that matched a construct barcode found in the reference file. The resulting log-normalized counts were used to generate the PCA plot.

8. Fig 4, panels E-G: (a). resolution is poor (canât see text in some axes or in the pathways)

8. Fig 4, panels E-G: (b). add legend for red/green/blue/black bar graph; also, numbers within bar graph are not readable.

Thanks for this feedback. We have changed this accordingly.

9. L302-329: Section on âPerturbation of environmental heme poolsâ : is hemin a chemical homolog for heme or for one of its precursors? The reason I ask is that I wonder if it is possible that the parasite doesnât grow in HMBS-deficient cells not because of the absence of heme per se but because the cells are damaged/deficient in a parallel pathway/molecule dependent on hydroxymethylbilane (or one of the other heme precursors). This question is resolved if hemin is replacing heme directly. Pls clarify.

The term âhemeâ refers to a protoporphyrin with an iron ion in its reduced state (Fe²⁺). This is usually reduced by proteins that bind this molecule (hemoglobin, cytochromes, etc.). However, free heme can be toxic, so the iron ion is quickly oxidized to Fe³⁺, and in this state this protoporphyrin is called âheminâ. Hence, hemin and heme represent the two natural forms (reduced/oxidized) of the same organic compound which are easily interchanged.

10. L50: Humans are animalsâ; Replace âin humans and animals.â with âin humans and other animals.â

-9-

Has been changed (see line 52)

11. L50-53: Since Plasmodium represents a whole group of species, it is not appropriate to refer to it as a âparasiteâ. Rephrase sentences to read: âMalaria-causing Plasmodium falciparum is the most impactful human parasite, with [â]. Other socio-economically important apicomplexans are the closely related livestock parasites in the genus Theileria, which kill more than 1 million (Mio) cattle each year and have ââ

Has been changed (see lines 52-58)

12. L70: Spell out complete species names the first time they are mentioned (*Plasmodium yoelii* <- notice typo in manuscript); L89-90: *Theileria annulata*
Has been changed (see lines 53, 80 and 99).

13. L165: show that SRM is → show that spermidine synthase (SRM) : I suggest spelling out gene name (especially since that is done in the rest of the sentence)
Has been changed (see line 188).

14. Be consistent throughout on italicizing genes but not their products (that seems in L165-167 as I assume that in L165 the authors are referring to the gene) but there are places where this breaks down (eg, L232-234).
We thank the reviewer for this remark, and we apologize for the lack of consistency. We have changed the text and the figures to refer to gene names in italics font and to protein names in regular font.

REVIEWER 3

The manuscript entitled "Host Cell CRISPR Genomics and Chemoinformatics Reveal Shared Metabolic Vulnerabilities in the Intracellular Development of *Plasmodium falciparum* and Related Hemoparasites" by Maurizio et al. describes an experimental and computational framework to understand host-parasite interactions during the infection *Plasmodium* and *Theileria*. The manuscript has impressive modeling work as well as experimental screenings of *Theileria*. Despite the efforts of the authors to connect the modeling outcomes of both host-pathogen interactions the workflow seems to have several missing parts in the *P. falciparum* site. If authors change the focus to *Theileria* rather than *Plasmodium* the story will be more comprehensive or in the worst-case scenario separate the two stories. Modeling of host-*P. falciparum* can be only an additional analysis. This new arrangement will help the organization of the manuscript because currently there are several jumps back and forth between the two systems. Lastly, testing of models architecture and simulations was not possible because models were missing in the submission and the lack of repository information.

We thank Reviewer 3 for the insightful comments on our manuscript. We appreciate the recognition of the impressive modelling work and experimental screenings performed in our study. The suggestion to potentially refine the focus by emphasizing *Theileria* rather than *Plasmodium*, or alternatively to present separate narratives for each parasite, is duly noted. We understand the importance of ensuring clarity and coherence in the organization of our manuscript, especially regarding the transitions between different systems. In addition, we acknowledge the concerns raised regarding the lack of model and repository information, which may have hindered a thorough evaluation of our work. In our response, we address each point raised by the reviewer and consider the best approach to improve the overall presentation and accessibility of our results.

- 10 -

Major comments:

The introduction of the concept parasitome is interesting, however host-pathogen genome scale metabolic models have been previously reconstructed (10.1038/msb.2010.68). Those models usually create a sub compartment that allows for exchange reactions among the host and parasite while enabling the flexibility to independently optimize for two or more biomass objective functions. Details about reaction reversibility and definition of the parasitome are missing or hard to find in the manuscript. It would be great if the authors add more details about this step.

We thank the reviewer for this insight, and we agree with the reviewer that the host-pathogen interactions can be modeled using both genome-scale models (GEMs) by creating a sub-compartment in the host GEM that contains the parasite GEM. However, in order to study the nutritional requirements of the parasite and to simplify the representation of the intracellular metabolic details of the parasite, we introduced the concept of the parasitosome: a unidirectional reaction representing the nutritional interactions of the parasite with the host. This reaction includes as substrates the nutrients that the parasite takes up from the host cell and as products the by-products that the parasite secretes into the host cell.

To clarify the generation of the parasitosome equation, we have added further details about the parasitosome steps in Figure 1D, clarified them in the main text and in the Methods section "The parasitosome as a representation of the metabolic state of *Plasmodium falciparum*", which now reads as follows in lines 668-695:

"The parasitosome is a unidirectional equation defined by the following steps:

1. Define the extracellular composition in the parasite model. In this case, we had two cases: (i) parasites in a auxotrophic mode, which take up the available nutrients from the host cytosol, (ii)

parasites in a (partially) prototrophic mode, which take up the minimum amount of nutrients from the host and use their intracellular metabolism to synthesize the rest of the compounds required for growth.

2. Use the lumpGEM method⁵² in the liver-iPfa model to identify the set of intracellular reactions that must be active to simulate parasite growth. This subnetwork is known as minimal network (MiN).

3. Then, lumpGEM generates a lumped reaction by collapsing the subnetwork into one reaction (see original publication for more details). In the resulting reaction, substrates represent the nutrients that the parasite takes up from the host's cytosol, and products represent the by-products that the parasite must secrete into the host's cytosol.

It is important to note that the subnetwork generated by lumpGEM contains the details of the metabolic reactions active for the corresponding metabolic configuration as well as the biomass reaction of the parasite. As a result, one of the metabolites in the parasitosome reaction is the parasite biomass. Therefore, the optimized flux through the parasitosome in the integrated model represents the overall growth of the parasite by uptake and secretion of compounds to and from the host cytosol. Note that the parasitosome can support growth up to the maximum value but it can also capture suboptimal growth.

The directionality of each parasitosome, as well as the directionality of the reactions composing the underlying subnetwork are determined by lumpGEM. The algorithm finds the optimal configuration of fluxes that maximize biomass production in the parasite model, taking into account the directionality and thermodynamic feasibility of the reactions as defined in the curated GEM30.

Note also that the alternative biochemistry of the parasite leads to alternative metabolic configurations (subnetworks) and thus to alternative parasitosomes.

Note that we have renamed the case studies to auxotrophic parasite and (partially) prototrophic parasite to design the previous cases of rich and compromised extra-parasitic environments, respectively.

- 11 -

The generation of the parasitome seems to have several limitations, for example the dependency on the fixed constraints (e.g. growth rate), which highly affects the feasibility of gene essentiality prediction. Another remarkable limitation is the optimization of a single biomass objective function. The lumpGEM method that authors previously published and implemented for this work enables the simulation of several physiological states, however algorithms such as OptCom are more broadly used. It would be interesting to make a comparison between the outcomes of both algorithms.

We thank the reviewer for this suggestion. In this case, the parasitomes are generated for maximum growth and we use the notion of the parasitome as the basis of all metabolic states that the parasites can achieve given specific environmental conditions (nutrients available in the cytosol of the host) and the parasite's biochemical capabilities (iPfa GEM). With the parasitome approach, we aim to understand what the important metabolic reactions in the system are by combining parsimony and flux analysis. We then identify host metabolic genes that are essential for parasites that grow operating in different metabolic states (represented as alternative parasitomes).

OptCom is designed to simulate interactions in a community, including the possibility of simulating parasitic interactions between organisms. OptCom simulates different organisms interacting through a common extracellular space. In our case, we have an intracellular parasite, and therefore, the OptCom code would need to be reformulated in order to capture the host-pathogen interactions, which are different from the community interactions as they do not share an extracellular environment. Moreover, in OptCom, one employs an objective for the "community", while in our case, such an objective is not needed, and we simply use another objective that identifies if the parasite, represented by the parasitome equation, can grow in the intracellular environment of the host cell. Furthermore, the studies presented in this work do not require bilevel optimization. With the formulation of the parasitome, we decouple the problem into two main parts, first, we study the nutritional requirements of the parasite by

- generating the alternative metabolic states of the parasite under certain conditions (minimal set of reactions required for parasite growth).

- identify the nutritional interactions with the host (uptake and secretion)

- use lumpGEM to formulate the interactions as a balanced chemical reaction, considering the parasite's metabolism as a single reaction.

In the second part, we investigate how the host can produce these nutrients required by the parasite, and what effect knocking out the genes that produce these nutrients might have on parasite growth (for the alternative metabolic states of parasites).

Minor comments:

Some introductory sentences of the discussion will be highly beneficial in the introduction

section. For example, diverse heme acquisition strategies have been described among Apicomplexan parasites. While *Plasmodium* and *Toxoplasma* possess a complete and functional de novo heme biosynthetic pathway, parasites such as *Babesia*, *Theileria* and *Cryptosporidium* depend on acquiring heme from the host cell.

We have thoroughly revised the manuscript accordingly and added a supplementary figure to clarify for the reader the differences between the host and parasite heme pathways among the apicomplexans of the study clearer for the reader.

The manuscript is missing a figure to connect both stories describing the methods and results. For example Fig. 1-4 focuses on *Plasmodium* while only Fig. 4 with a totally different format

- 12 -

focuses on *Theileria* data.] [Figure 5 should be expanded to account for the results of *Theileria*. We thank the reviewer for this feedback, which is consistent with reviewer 1. Following the suggestions of reviewers #1 and #3, we have now added an improved figure (Figure 1A) that summarizes the rationale behind our choice to focus on *Plasmodium falciparum* and *Theileria annulata*. It also shows the currently feasible screening technologies to identify key metabolic host genes for the growth of both apicomplexan parasites. In addition, Figure 5 connects the findings and highlights the overlap between the *Theileria* metabolic essentialome and *P. falciparum* essentiality predictions, focussing not only on the metabolic pathways but also on the corresponding essential host genes.

Line 126. A clarification if the reduction of 165 to 47 nutrients is based on experimental evidence is missing.

We thank the reviewer for this comment, and we apologize for the lack of clarity. The selection of nutrients is based on the assumption that the parasite (*P. falciparum*) uses nutrients available in the cytosol of the host cell (human hepatocyte). Therefore, 165 nutrients were identified as the overlap between metabolites from the extracellular compartment of the *P. falciparum* genome-scale metabolic model (iPfa) and the metabolites available in the cytosol of the hepatocyte model. Aiming to investigate the metabolic capabilities of *P. falciparum* to survive inside the hepatocyte, we simulated two cases:

1. A parasite that has access to all the metabolites in the host cytosol and takes them up to produce biomass precursors. Initially, we referred to this case study as a parasite growing in a rich medium. However, after the feedback from this review, we have decided to rename it to a parasite growing in auxotrophic mode. This study simulates the case where the parasite uses the nutrients from the host cytosol, even if it can synthesize them.

2. A parasite that, although it has access to all the metabolites in the host cytosol, uses only the minimum required from the host (the minimum required for the parasite to grow) and synthesizes all the rest. In this case, we used the *P. falciparum* genome-scale metabolic model (iPfa) to identify the minimum nutrients that the parasite needs to take up from its extra-parasitic space to produce biomass. We used the iMM method and obtained 47 nutrients required by the parasite. In our first submitted manuscript, we referred to this case study as a parasite growing in a compromised medium. However, we renamed it to a parasite growing in (partially) prototrophic mode, where the parasite is able to synthesize some of the required biomass precursors even if they are available in the host cytosol.

These two studies aim to identify the extreme metabolic states that the parasites can acquire to survive inside the host cell. Moreover, to identify host genes as good targets to stop parasite growth, these two case studies cover the possibility of targeting a gene that may seem essential when the parasite is residing in an auxotrophic mode, but in reality the parasite may have the ability to rewire its metabolism towards a (partial) prototrophic mode and synthesize the required metabolite on its own, leading to a failed target for parasite survival.

We have further clarified this in the Results section in lines 137-152:

One must consider the parasite's ability to adapt its metabolism within the host cell to identify effective host gene targets for inhibiting parasite proliferation. For instance, *P. falciparum* demonstrates versatility by scavenging some nutrients from the host cytosol (as an auxotrophic parasite) or synthesizing them when unavailable (as a partial prototrophic parasite). This adaptability is crucial to understand when designing compounds that target host enzymes responsible for nutrient production. Therefore, we used the liver-iPfa model to investigate both scenarios: when *P. falciparum* operates in an auxotrophic mode or in a partial prototrophic mode (Fig. 2A). In the first case (auxotrophic), we simulated a parasite that has access to metabolites present in the cytosol of the hepatocyte model, resulting in 165 nutrients (Table S1 and Methods).

- 13 -

In the second case (partial prototrophic), we simulated a parasite that only requires to consume the minimal number of metabolites from the hepatocyte cytosol for survival, resulting in a set of 47 nutrients (Fig. S1A, Table S1 and Methods). The reduction in nutrient requirements in the partial prototrophic case is explained by the fact that the parasite synthesizes some of the metabolites that

it takes up from the host in the auxotrophic case.â

Fig.2. Please clarify if any of the parasitomes and selected pathways was selected based on experimental data or only simulations.

We thank the reviewer for raising this question. All the parasitomes in this work were generated based on simulations. However, it is worth noting that the GEM of iPfa was reconstructed based on experimental evidence of metabolic enzymes expressed in *P. falciparum* and its ability to grow on different media conditions (doi:10.1371/journal.pcbi.1005397). Therefore, the biochemical capabilities of the GEM and the biomass composition are experimentally tested. What is hypothetical is the accessibility to the nutrients in the host, which will partially define the metabolic state of the parasite.

Fig. 4D. Please use the models to perform this network analysis and move the current figure to supplementary materials

The reviewer has requested that we utilize the network analysis model and move the current figure to the Supplementary Material. The exact instructions are unclear to us. We believe it would be unfortunate to relegate STRING to supplementary figures, as we find it highly informative for the reader. Therefore, if the editor agrees, we suggest keeping the figure as it is.

Insights about how models are going to be made publicly available is missing. Please describe the formats and repositories in which the models, code and workspaces will be shared.

We thank the reviewer for pointing this out, and we apologize for this unintentional omission of the information. We had already created a public GitHub repository containing all the data, models, and code required to reproduce the results of our modeling study of the metabolic interactions between *P. falciparum* and the human hepatocyte but we failed to refer to this during submission. We have now added this information to the Data and Code Availability sections.

REVIEWER 4

The goal of this study was to identify shared vulnerabilities for intracellular apicomplexan parasite development. The authors used a novel hepatocyte metabolic reconstruction to investigate dependency of *Plasmodium falciparum* on liver metabolism. This approach took advantage of tools previously developed by the authors including a parasite metabolic reconstruction (iPa) and a method to generate subnetworks (LumpGem) as well as a clever exploration of rich vs compromised conditions. The authors then assessed the effects of a CRISPR KO screen to explore the metabolic dependencies of a related apicomplexan parasite (*Theileria*) in its host macrophage. The top hits were validated using biochemical inhibitors. Together, the authors identify a core set of host metabolic genes required for parasite survival including purine and heme biosynthesis. However, an overall lack of rationale and discussion of chosen approaches limit the potential for broader appreciation, and thus significance; much more description is required to be useful for the community. Also, the major biological conclusions from these studies are modest given that it is well established that purine salvage is a major parasite weakness. The conclusions do contribute a viewpoint to the growing story about heme salvage requirements but do not finalize this story.

- 14 -

We thank Reviewer 4 for the insightful feedback on our manuscript. The acknowledgement of our novel approach using hepatocyte metabolic reconstruction to investigate *Plasmodium falciparum* dependence on liver metabolism, is duly noted. The reviewer's recognition of our clever exploration of rich vs. compromised conditions and subsequent evaluation of CRISPR KO screens to explore metabolic dependencies in *Theileria* is appreciated. We understand the concern regarding the lack of rationale and discussion, and we acknowledge the need for more detailed descriptions to increase the usefulness of our findings to the community. In our response, we have addressed each of the points to improve the clarity and impact of our manuscript.

Major comments:

1. There are numerous new models and concepts present in the manuscript that are not validated or are missing rationale or justification. Addressing these "sticking points" are critical to ensure that the broader audience appreciates the choices that the authors have made during their approach. For example:

- The manuscript describes a novel GEM of hepatocyte metabolism but does not present any validation that this model is useful and accurate. Entire papers explore the creation and validation of models following standards of the field.

We thank the reviewer for raising this concern. The hepatocyte and the host-parasite models

generated in this work passed the generic metabolic tasks defined in the field for other human GEMs such as Human1 and Recon 3D. We have explicitly included these results in Supplementary Figure S10. In addition, we have also included the MEMOTE score obtained for our generated model, which is another well-established standard for GEM quality.

- A new concept in the paper is the idea of the "parasitosome", which are defined using a tool the authors developed previously (LumpGEM). This is the first time the tool has been applied to the parasite/intracellular context and the authors do not describe the rationale behind using the parasitosome concept in the context of the liver.

We thank the reviewer for this insight. We have subsequently revised both the main text and the Methods section entitled "The parasitosome as a representation of the metabolic state of *Plasmodium falciparum*" to provide enhanced motivation and to describe the parasitosome reactions further.

- Reconstructing the metabolism of an intracellular parasite is inherently complex and requires many assumptions. The manuscript lacks details that justify the chosen approach involving mapping "extraparasitic metabolites" to the liver cytosol. The attempt to explain the approach (line 552) is unclear while Fig 1 legend is somewhat better. Is there precedence in the literature that justify this route?

We thank the reviewer for this comment and we apologize for the lack of clarity. The selection of extra-parasitic metabolites is based on the assumption that the parasite has access to nutrients available in the cytosol of the hepatocyte. To model this, we had to identify the extracellular metabolites from the parasite model that are present in the cytosol of the hepatocyte model, as these are the nutrients that the parasite will be able to take up. Having done that, we studied different metabolic configurations of the parasite when it uses all the nutrients from the extracellular space "auxotrophic parasite" or the minimum it needs for growth and synthesizes the rest by itself "partial prototrophic parasite".

We have rephrased this in the main text (lines 137-152) to indicate the two scenarios we are studying to explore the complex metabolism of the parasite:

"One must consider the parasite's ability to adapt its metabolism within the host cell to identify

- 15 -

effective host gene targets for inhibiting parasite proliferation. For instance, *P. falciparum* demonstrates versatility by scavenging some nutrients from the host cytosol (as a auxotrophic parasite) or synthesizing them when unavailable (as a partial prototrophic parasite). This adaptability is crucial to understand when designing compounds that target host enzymes responsible for nutrient production. Therefore, we used the liver-iPfa model to investigate both scenarios: when *P. falciparum* operates in a auxotrophic mode or in a partial prototrophic mode (Fig. 2A). In the first case (auxotrophic), we simulated a parasite that has access to metabolites present in the cytosol of the hepatocyte model, resulting in 165 nutrients (Table S1 and Methods). In the second case (partial prototrophic), we simulated a parasite that only requires to consume the minimal number of metabolites from the hepatocyte cytosol for survival, resulting in a set of 47 nutrients (Fig. S1A, Table S1 and Methods). The reduction in nutrient requirements in the partial prototrophic case is explained by the fact that the parasite synthesizes some of the metabolites that it takes up from the host in the auxotrophic case."

- While the comparison of rich vs compromised conditions is a useful comparison to explore the metabolic capabilities of the parasite, the manuscript is missing any physiological justification of these conditions.

We thank the reviewer for bringing this concern to their attention. If we assume that the parasite has access to all nutrients, we neglect its ability to synthesize compounds on its own. Consequently, we may incorrectly identify host genes as essential for the parasite, when in fact the parasite can adapt its metabolism to synthesize it and survive without them being supplied by the host. For instance, a metabolite may be considered essential in a nutrient-rich medium, but if the parasite can synthesize it, targeting it with drugs may not be effective. The IMM analysis helps to identify the minimum number of nutrients required for parasite growth. As a result, genes identified as essential by this analysis are more likely to be genuinely indispensable for the parasite, as it narrows down the focus to genes that the parasite really needs.

To have a more physiological description of our analysis, we have renamed the two case studies to parasite acting in a auxotrophic mode (previously named parasite in a rich environment), when it scavenges all the nutrients from the host's cytosol, and parasite in a partial prototrophic mode (previously called parasite in a compromised environment), which simulates a parasite that scavenges the minimum required metabolites from the host cytosol and uses its internal biochemistry to synthesize the rest of its biomass precursors.

2. The manuscript is missing a discussion of the limitations of the bovine macrophage system. It is left to the reader to piece together that two different cell lines must be used as uninfected and infected systems (BoMAC and TaCl2- this information is not defined until the Methods section,

see minor point below). Perhaps there is a reason behind this (i.e. cell line availability). However, comparison of these different cell lines leaves the door open that differences in essential genes are due to host cell lineage and not solely infection status. The manuscript would benefit from more details on this.

We thank the reviewer for pointing this out. We have now added details and a discussion of the limitations of the bovine macrophage system in lines 240-243 as follows:

“BoMac cells, derived from a bovine monocyte/macrophage lineage, represent the most effective system currently available for assessing the essentiality of bovine host genes within a macrophage background”.

We acknowledge that comparing different cell lines, such as BoMac and TaC12, albeit both of macrophage lineage, is not ideal; however, for technical reasons, it is currently the best option available in our experience. Conducting a genome-wide dropout CRISPR screen necessitates a proliferating cell line, ruling out primary bovine macrophages for *Theileria* infection. Additionally, the tropism of *T. annulata* is confined to bovine B cells and primary bovine macrophages, limiting

- 16 -

available cell lines for in vitro studies and ruling out BoMacs. Although immortalized bovine B cells (BL20 cell line) infected with *T. annulata* (TBL20 cell line) are available, BL20 cells are refractory to transduction, as also reported by others (Serafini et al., 2004; Janssens et al., 2003; Bovia et al., 2003), precluding the use of the BL20/TBL20 pair for a genome-wide dropout screen.

3. Again, the lack of rationale across the paper opens major questions for this reviewer. The computational approach using the integrated hepatocyte-*Plasmodium* model is distinct from the “*Theileria* essentialome” approach during the CRISPR screen, yet they are directly compared at the end.

The revised Figure 1 now provides the rationale for the multidisciplinary approach taken in this project. Experimental limitations on the liver stage of *P. falciparum* and limited omics data for *Theileria* limit the ability to study host-parasite interactions on a large scale using identical technology. We have therefore decided to study the host metabolic requirements of these pathogens on a large scale using the most appropriate tools. In lines 209-213 we now write:

“Having identified essential host genes for survival of *P. falciparum*, we next aimed at finding common host factors essential for *T. annulata*. While the generation of the hepatocyte-*P. falciparum* metabolic model was made possible by recent advances in modeling resources 26-31, omics data for bovine leukocytes and *Theileria* parasites remain limited and incomplete”. Please see also our response to a similar comment from reviewer #1.

- In the first half of the paper, the authors identify genes essential to parasite, yet dispensable to hepatocyte. This point is clearly described, but not justified (and convoluted across the paper- see Figure 5).

We thank the reviewer for this remark. Our objective was to target proteins crucial for the parasite’s function while being non-essential for the host cell. This approach aims to identify potential drug targets to impede parasite growth without harming the host cell. In the revised manuscript, we have clarified this in lines 107-109:

“Aiming to understand the metabolic dependency of the malaria parasite *P. falciparum* during the liver stage, we developed a systems biology approach to explore how the parasite acquires nutrients from the host.”

and in lines 131-133:

Using this approach, we investigated how the essentiality of host genes for parasite survival changes across different metabolic configurations of *P. falciparum*, identifying potential drug targets.”

-Then in the second half of the paper, the authors take essential genes from one cell line (BoMacs) and subtract these from list of essential genes from a different cell line (TaC12) that is infected with *Theileria*. To this reviewer, this comparison yields genes that are essential for the macrophage when *Theileria* is around. The authors again leap from “genes essential in *Theileria*-infected macrophage- to “*Theileria* essentialome” (line 214), which is misleading and unjustified. We acknowledge the concern raised by the reviewer. To provide clarification, we categorize the hits obtained in the infected macrophage (TaC12) screen into three main groups:

1) Host (bovine macrophage) essential genes: Each cell line or lineage has a core set of genes that are essential for survival. Different cell types have different essential gene sets, although there is some overlap (e.g. genes encoding tRNAs). Ideally, this set of genes should be common to different bovine monocyte/macrophage cell lines (BoMac, TaC12). Therefore, we do not consider the exclusion of BoMac essential genes from the TaC12 hit list to be a significant problem or an inappropriate practice.

2) *Theileria* survival host factors: These are host genes specifically required for schizont survival but dispensable for the viability of uninfected bovine macrophages. Identification of these factors was a primary goal of our screen. The concept is that knockout (KO) of a host factor essential for *Theileria*

- 17 -

survival renders the parasite non-viable. As a result, the cancerous phenotype induced by *Theileria* in the host cell is lost and the cells do not progress to the end of the dropout screen. Instead, they revert to a dormant phenotype, resulting in a significant depletion of the KO gene in the pool. We have added lines 218-220 to clarify this: "Survival and growth of the transformed cell depends upon the presence of a viable parasite³², and so loss of host cell viability will in many cases indicate a loss of parasite survival."

3) *Theileria* transformation factors: These are host genes necessary to maintain or induce the transforming state of the host cell, characterized by continuous proliferation induced by the parasite. Similar to *Theileria* survival factors, the inactivation of a gene critical for hyperproliferation of the cell by Cas9 results in loss of continuous proliferation.

Regarding the second part of the comment, we have defined the term "*Theileria* essentialome" in the main text (lines 249-252) and it serves as a concise version of "*Theileria*-infected macrophage essentialome" and facilitates its use throughout the text.

- Comparing essential genes for the parasite to essential genes for the macrophage is less worthwhile than comparing genes essential to host cells -hepatocyte to macrophage- or parasites -Plasmodium to *Theileria*.

We thank the reviewer for raising this point, which is a similar concern as raised by reviewer 1. As mentioned in response to reviewer #1's comments, the access to omics data is insufficient to generate a metabolic model of a *Theileria*-infected bovine macrophage or even an uninfected bovine macrophage. Consequently, our best approach to investigate essential host factors for this parasite was through CRISPR screening. The identification of parasite-specific host factors requires the filtering of hits from the screen in uninfected macrophages. Approximately 800 genes from the sublibrary were scored in both the TaC12 and BoMac screens, which we classified as core essential genes for the bovine monocyte/macrophage lineage. However, it's worth noting that the model-predicted essential genes for human hepatocytes are primarily metabolic genes, whereas these 800 hits include genes involved in various other biological processes. Therefore, comparing essential genes for host cells (hepatocyte vs. macrophage) may not be as informative as originally expected.

Minor comments:

1. Line 82: Over the past several years, metabolic reconstructions have become an exciting way to explore parasite metabolism. Many references are missing here.

The authors thank the reviewer for this comment, and we have added the following sentence and references to the Introduction (lines 89-91): "In recent years, GEMs have been instrumental in investigating parasite metabolism^{20,21}, including that of apicomplexan parasites^{10,22-25}."

2. Line 126: The manuscript is missing justification for the two scenarios.

We thank the reviewer for raising this concern. To have a more physiological description of our analysis, we have renamed the two case studies to parasite acting in a auxotrophic mode (previously named parasite in a rich environment), when it scavenges all the nutrients from the host's cytosol, and parasite in an prototrophic mode (previously called parasite in a compromised environment), which simulates a parasite that scavenges the minimum required metabolites from the host cytosol and uses its internal biochemistry to synthesize the rest of its biomass precursors.

We have added the following paragraph in the Results section to justify the analyses performed in this work in lines 137-144:

"One must consider the parasite's ability to adapt its metabolism within the host cell to identify effective host gene targets for inhibiting parasite proliferation. For instance, *P. falciparum* demonstrates versatility by scavenging some nutrients from the host cytosol (as a auxotrophic - 18 -

parasite) or synthesizing them when unavailable (as a partial prototrophic parasite). This adaptability is crucial to understand when designing compounds that target host enzymes responsible for nutrient production. Therefore, we used the liver-iPfa model to investigate both scenarios: when *P. falciparum* operates in a auxotrophic mode or in a partial prototrophic mode (Fig. 2A)."

3. The reference to "case study I/II" in figures is not consistent with text (line 126, called "scenarios")

We thank the reviewer for pointing this out. We have now corrected the reference to the case studies.

4. Line 134: should the figure reference be Fig S1C-D?

We thank the reviewer for bringing this up. We have now corrected the reference in the main text (lines 154, 157 and 159). Figure S1 has been modified. The scheme that was before Figure S1A has been moved now to the main figure (Fig. 1C). Therefore, the updated references are now Fig. S1B

and Fig. S1C.

5. Line 191: rephrase to "500x coverage of the genome"

We have rephrased the sentence in lines 224-226 as follows: "The genome-wide screen was performed at a coverage where each sgRNA is expressed in at least 500 cells and in three biological replicates."

6. Line 210 "About 80% of candidate genes were validated as crucial for parasite survival (Fig. S4F)." -What candidate genes? From what cell line? The total 1500 tested? Be clearer here.

We thank the reviewer for highlighting this. We have revised the text to provide clearer details. In lines 248-249 we now write "About 80% of the ~1,500 candidate genes identified in TaC12 cells were validated as crucial for parasite survival."

7. Line 233: "While enzymes that convert IMP to AMP are present (ADSS and ADSL), those deputed to synthesize GMP (IMPDH and GMPS) are absent." -need more clarity here: absent/present from pathways, or essentiality lists?

Before the quoted sentence, the text mentions that the entire phylum of apicomplexan parasites lacks the de novo purine biosynthetic pathway (lines 266-267). It then explains that the *Theileria* essentialome, as expected, includes numerous host genes involved in purine precursor biosynthesis. For clarification, the term "host" has been added to specify that we are discussing host genes that are essential for *Theileria*-infected macrophages. Therefore, the genes encoding the enzymes ADSS/ADSL/IMPDH/GMPS are either present or absent in the *Theileria* essentialome, which the reviewer refers to as the "essentiality list".

8. Line 238: need to define GSS and GCLC before reference in Results.

Has been changed (lines 274-275).

9. Line 266: FECH definition

Has been changed (line 305).

10. Line 279: How do the authors justify 99 chosen enzymes? Stated earlier that 653 essential genes defined as "essentialome", 15% were metabolic- make this point clearer

We appreciate the reviewer's question. A total of 653 essential genes were identified for *Theileria*-infected macrophages. These genes were classified into metabolic and non-metabolic categories - 19 -

using GEM. As stated in the text (line 320), 99 of the 653 essential genes encode metabolic enzymes.

11. Fig 4/S4, Panel A- choice of the two greens (sage and neon) are confusing and cannot be distinguished by this reviewer.

We have have changed one color to black.

12. The manuscript would benefit from some additional biological details including pathways for heme and glutathione biosynthesis to better convey how these pathways are used in the parasites.

We thank the reviewer for the suggestion. We have included the pathways for heme and glutathion biosynthesis of the parasites in the Supplement (Figure S9) and we refer to this figure in lines 342-344: "The prediction that host HMBS is essential for *P. falciparum* was completely unexpected, since *Plasmodium* possesses the entire enzymatic machinery for heme biosynthesis (Fig. S9)."

Version 2:

Reviewer comments:

Reviewer #1

(Remarks to the Author)

The authors have addressed my concerns in their revised, and improved, manuscript.

Reviewer #3

(Remarks to the Author)

NCOMMS-23-52726A-Z-R1

The authors addressed all my comments and suggestions. However, after looking at the documentation and model simulations it seems like the parasitome approach is groundbreaking and can potentially change how host-pathogen interactions are simulated at genome-scale. As the authors acknowledge OptCom has been previously used to simulate

parasitic interactions. Thus, a comparison between both approaches is highly interesting. I suggest the authors incorporate this comparison. As the authors demonstrated that the parasitome will unravel essential genes for parasites based on different metabolic stages and I expect that in comparison with OptCom not many essential genes will be predicted. A comparative table of predicted gene essentiality will demonstrate that the parasitome approach is the most accurate to simulate host-pathogen interactions.

If predictive host-pathogen computational tools such as the parasitome are to be broadly adopted by the scientific community an explicit comparison with current tools is necessary. This system(s) seems to be an ideal system to demonstrate the predictive power of the parasitome because of all the molecular biology data (e.g. CRISPR/Cas9).

Minor

Line 135 Missing capital in "In silico"

Fig. 5. Please mark what genes were experimentally validated.

Reviewer #4

(Remarks to the Author)

The authors have sufficiently addressed the concerns of this reviewer.

Author Rebuttal letter:

We thank the editor and the reviewers for the positive feedback. Please find our point-by-point response in the Author Checklist and below. We have again highlighted the changes in the manuscript.

REVIEWERS' COMMENTS

REVIEWER 3

The authors addressed all my comments and suggestions. However, after looking at the documentation and model simulations it seems like the parasitome approach is groundbreaking and can potentially change how host-pathogen interactions are simulated at genome-scale. As the authors acknowledge OptCom has been previously used to simulate parasitic interactions. Thus, a comparison between both approaches is highly interesting. I suggest the authors incorporate this comparison. As the authors demonstrated that the parasitome will unravel essential genes for parasites based on different metabolic stages and I expect that in comparison with OptCom not many essential genes will be predicted. A comparative table of predicted gene essentiality will demonstrate that the parasitome approach is the most accurate to simulate host-pathogen interactions. If predictive host-pathogen computational tools such as the parasitome are to be broadly adopted by the scientific community an explicit comparison with current tools is necessary. This system(s) seems to be an ideal system to demonstrate the predictive power of the parasitome because of all the molecular biology data (e.g. CRISPR/Cas9).

We agree with the editors that comparing our approach to OptCom is beyond the scope of this work. Based on our ongoing work on methods development in this research area, we would like to offer the following arguments in support of this. There are important differences in the formulation of the OptCom optimization framework and the approach we propose in this paper, such as the presence of inner optimization problems embedded in an outer optimization problem and the type of interactions considered (intracellular interactions vs. interactions mediated by an extracellular environment). Based on our ongoing investigation of these methods, we believe adapting the OptCom formalism to study intracellular host-parasite interactions and our current method to other organisms and communities is feasible. However, we also believe that such work is best suited to a methods paper and is, therefore, beyond the scope of our current paper. In addition, from our preliminary investigation, we can conclude that gene essentiality predictions obtained from a revised formulation of OptCom would likely yield more essential host genes, mostly false positive predictions, than the parasitome approach. This is because the predictions from our approach are based on more stringent parsimonious constraints regarding the minimal sets of substrates (for the partially prototrophic case) and the minimal sets of intracellular reactions (minimal networks) required for the parasitome equations to produce biomass. The predictions from our approach will have a higher confidence score and will also be found as a subset of the predictions obtained from the OptCom approach. Additionally, we confirm that the code was written to run on IBM's free community version of CPLEX. Nevertheless, we have conducted testing with Gurobi as a solver and have incorporated this option into the code. Currently, users have the flexibility to choose between CPLEX or Gurobi. The code is publicly available at https://github.com/EPFLCSB/host_parasite_interactions.

Minor

Line 135 Missing capital in "In silico"

Has been corrected.

Fig. 5. Please mark what genes were experimentally validated.
We marked the genes in the figure and added a sentence to the legend indicating in which parasites the genes were validated.

-2-

Please write a case for Gurobi to test the code.

We have conducted testing with Gurobi as a solver and have incorporated this option into the code. Currently, users have the flexibility to choose between the IBM's free community version of CPLEX or Gurobi.

The code is publicly available at https://github.com/EPFL-LCSB/host_parasite_interactions.
